# Stratospheric influence on North Atlantic marine cold air outbreaks following sudden stratospheric warming events

Hilla Afargan-Gerstman[1], Iuliia Polkova[2], Lukas Papritz[1], Paolo Ruggieri[3,4], Martin P. King[5],
Panos J. Athanasiadis[4], Johanna Baehr[2], and Daniela I.V. Domeisen[1]

[1]Institute for Atmospheric and Climate Science, ETH Zürich, Zürich, Switzerland
[2]Institute of Oceanography, Universität Hamburg, CEN, Hamburg, Germany
[3]Department of Physics and Astronomy, University of Bologna, Italy
[4]Euro-Mediterranean Center on Climate Change (CMCC), Bologna, Italy
[5]NORCE Climate, and Bjerknes Centre for Climate Research, Bergen, Norway

**Correspondence:** Hilla Afargan-Gerstman (hilla.gerstman@env.ethz.ch)

**Abstract.** Marine cold air outbreaks (MCAOs) in the northeastern North Atlantic occur due to the advection of extremely cold air over an ice-free ocean. MCAOs are associated with a range of severe weather phenomena, such as polar lows, strong surface winds and intense cooling of the ocean surface. Given these extreme impacts, the identification of precursors of MCAOs is crucial for improved long-range prediction of associated impacts on Arctic infrastructure and human lives. MCAO frequency

has been linked to the strength of the stratospheric polar vortex, but the study of connections to the occurrence of extreme stratospheric events, known as sudden stratospheric warmings (SSWs), has been limited to cold extremes over land. Here, the influence of SSW events on MCAOs over the North Atlantic ocean is studied using reanalysis datasets. Overall, SSW events are found to be associated with more frequent MCAOs in the Barents Sea and the Norwegian Sea compared to climatology, and less frequent MCAOs in the Labrador Sea. In particular, SSW events project onto an anomalous dipole pattern of 500-hPa

geopotential height, which consists of a ridge anomaly over Greenland and a trough anomaly over Scandinavia. By affecting the variability of the large-scale circulation in the North Atlantic, SSW events contribute to the strong northerly flow over the Barents and Norwegian Seas, and thereby increase the likelihood of MCAOs in these regions. In contrast, the positive geopotential height anomaly over Greenland reduces the probability of MCAOs in the Labrador Sea after SSW events. As SSW events tend to have a long-term influence on surface weather, these results are expected to benefit the predictability of

MCAOs in the Nordic Seas for winters with SSW events.

## 1 Introduction

Marine cold air outbreaks (MCAOs) in the North Atlantic are characterized by large-scale advection of cold polar air masses over open ocean. These events are associated with a range of severe weather phenomena, such as polar lows (e.g., Mansfield, 1974; Rasmussen et al., 2004; Kolstad, 2011; Mallet et al., 2013; Radovan et al., 2019; Terpstra et al., 2016), strong surface

winds (Kolstad, 2017), and intense ocean-atmosphere heat exchange, playing an important role for deep-water formation (e.g.,

Isachsen et al., 2013; Buckley and Marshall, 2016). MCAOs occur throughout the year, however they are most frequent in the Northern Hemisphere in winter (Fletcher et al., 2016).

MCAOs are typically embedded in the northerly or northwesterly flow found in the cold sectors of midlatitude cyclones (Kolstad et al., 2009; Jahnke-Bornemann and Brümmer, 2009; Fletcher et al., 2016; Papritz and Grams, 2018; Pithan et al.,
2018). Consequently, variations in the frequency of MCAOs are closely related to variations in the frequency of cyclones slightly east (Papritz and Grams, 2018). Particularly preferred conditions for MCAOs in the northeastern North Atlantic occur during cyclonically dominated weather regimes, such as the Scandinavian trough and Atlantic ridge regimes that are associated with enhanced cyclone frequency in the Norwegian Sea and the Barents Sea, respectively (Papritz and Grams, 2018). In contrast, blocked regimes with anticyclonic flow anomalies over northern Europe (such as European or Scandinavian blocking)
tend to suppress cyclone activity in the Nordic Seas and, therefore, lead to a reduced MCAO occurrence in these regions. An important exception is Greenland blocking, which favours cyclone occurrence in the Barents Sea and is accompanied by frequent MCAOs in the Norwegian and western Barents Seas. A similar relation exists also for cyclone activity in the Irminger Sea and MCAOs in the Labrador Sea (Fletcher et al., 2016). In addition to storm track activity, the propagation of tropopause polar vortices, which are associated with a dome of extremely cold air masses (Cavallo and Hakim, 2010), has been found to
be an important mechanism for inducing the most intense MCAOs south of Fram Strait (Papritz et al., 2019).

The stratosphere can have a significant impact on surface weather in the North Atlantic and Europe. This has dominantly been investigated with respect to cold temperature extremes over land, where cold air outbreak frequency has been linked to the strength of the stratospheric polar vortex (Thompson et al., 2002; Kolstad et al., 2010; King et al., 2019). Extreme changes in the stratospheric polar vortex, such as sudden stratospheric warming (SSW) events, often lead to a negative signature of the North
Atlantic Oscillation (NAO; e.g., Baldwin and Dunkerton, 2001; Limpasuvan et al., 2004; Butler et al., 2017; Charlton-Perez et al., 2018; Domeisen, 2019), though with large tropospheric differences between events (Afargan-Gerstman and Domeisen, 2020; Domeisen et al., 2020). The tropospheric anomalies after SSW events can persist for up to two months, thus providing a key for improved predictive skill of surface weather on sub-seasonal to seasonal time scales (Sigmond et al., 2013; Scaife et al., 2016; Domeisen et al., 2019). Over Europe, weak vortex events are associated with up to 50% more cold days compared
to climatology (Kolstad et al., 2010), while about 60% of the observed cold temperature extremes in midlatitude Eurasia since 1990 can be explained by weak stratospheric polar vortex states (Kretschmer et al., 2018).

Since the previously discussed dominant pressure dipole pattern favouring MCAOs over the Nordic Seas is relatively independent of the NAO (e.g., Jahnke-Bornemann and Brümmer, 2009), the question arises to what extent the modulation of North Atlantic surface weather by stratospheric variability affects MCAO frequency. While the frequency of cold extremes over land
has been shown to increase in response to a weak stratospheric polar vortex, Papritz and Grams (2018) found indications that for winters with a weak polar vortex, MCAOs are in fact less frequent in certain regions, namely the Greenland and Iceland Seas due to a weakening of the storm track in the Norwegian Sea, whereas in the Barents Sea a modest increase of MCAO frequency was found. This suggests a spatially complex, not yet fully understood linkage between stratospheric variability and MCAO formation.

Here, the role of stratospheric extreme events in setting favorable conditions for MCAO occurrence over the North Atlantic is revisited using atmospheric reanalysis. Particularly, we focus on the Norwegian and Barents Seas, where the occurrence of polar lows is frequent and poses a risk on the increasing marine activity in these regions, and along the fairly densely populated Norwegian coast. In addition, the Barents Sea is the region with the most dramatic changes associated with Arctic amplification (Cohen et al., 2014). Another region of interest is the Labrador Sea, where MCAOs and their associated heat and moisture fluxes from the ocean into the atmosphere may have an impact on dense water formation in the North Atlantic (Gebbie and Huybers, 2010). A better prediction of MCAOs in these regions, due to the increased persistence of particular surface impacts after SSW events (as compared to climatology) would therefore be societally and economically beneficial. Our results aim to shed light on the precursors and occurrence of MCAOs over the Barents and Norwegian Seas, as well as for the Labrador Sea, which is expected to benefit long-range predictability of their extreme impacts.

## 2 Data and Methodology

### 2.1 Data

We use daily reanalysis data from the European Centre for Medium-Range Weather Forecasts (ECMWF) ERA-Interim dataset (Dee et al., 2011) from 1 January 1979 to 31 August 2019. The ERA-Interim variables examined include skin temperature (SKT), sea level pressure, 850-hPa temperature (T850), 10-m meridional wind, as well as 500-hPa zonal and meridional wind, and geopotential height (Z500) fields. Climatology is defined using long-term means of daily averages for this period (1979–2019), and daily anomalies are computed as deviations from long-term means. For storm track activity we use cyclone frequencies derived from sea level pressure with the method by Wernli and Schwierz (2006). A similar approach has been implemented in Papritz and Grams (2018). We focus on extended winter season from December to March (DJFM).

### 2.2 MCAO index definition

We use the MCAO index (M) for the classification of marine cold air outbreaks. The MCAO index was designed to detect the flow of cold air over a warmer ocean (Kolstad and Bracegirdle, 2008; Papritz et al., 2015; Polkova et al., 2019) and is defined as

$$M = \theta_{\mathrm{skt}} - \theta_{\mathrm{850hPa}} \tag{1}$$

where $\theta_{\mathrm{skt}}$ is the potential temperature at the ocean surface, computed from skin temperature and surface pressure. $\theta_{\mathrm{850hPa}}$ is the potential temperature at 850 hPa. The MCAO index is defined only over the ocean, thus land grid points are masked. Only positive potential temperature differences ($\theta_{\mathrm{skt}}$-$\theta_{\mathrm{850hPa}} > 0$ K), which indicate a state of atmospheric instability associated with upward sensible heat fluxes, are considered. In this study we focus on MCAOs with an MCAO index (Eq. 1) exceeding a threshold of 4 K. This choice of threshold is in accordance with the thresholds used in Papritz and Spengler (2017) and selects moderate-to-strong MCAOs with notable upward heat fluxes from the ocean. Other studies have used slightly different

thresholds for the MCAO index (e.g., a threshold of 3 K for moderate MCAO events in Fletcher et al. (2016)), however, the results are not sensitive to small changes of this threshold (on the order of 1-2 K).

## 2.3    Characterization of the large-scale flow

We define a new index based on the 500-hPa geopotential height anomaly from climatology ($Z'$). The index, named the Zonal Dipole Index (ZDI), is equal to half of the difference in $Z'$ between the spatial average over two main areas that modulate the

frequency of MCAOs in the Barents and Norwegian Seas (enclosed by the green boxes in Fig. 2d): southeast of Greenland ($Z'_G$, 30°W-50°W, 60°N-70°N) and northern Europe ($Z'_E$, 30°E-50°E, 60°N-70°N), as follows

$$\text{ZDI} = \frac{Z'_G - Z'_E}{2}. \tag{2}$$

## 2.4    SSW events

To assess the impact of the stratosphere on MCAO occurrence, we examine the changes in the MCAO frequency in response

to 26 observed major SSW events between 1979–2019. Major SSWs occur when the westerlies associated with the winter stratospheric polar vortex reverse to easterlies. A common definition for the central date of major SSWs is based on a change from westerly to easterly of the daily mean zonal-mean zonal winds at 10 hPa and 60°N between November and March (Charlton and Polvani, 2007). A list of major SSW events in the ERA-Interim reanalysis for the period 1979–2019 is shown in Table 1. The central dates of SSW events between 1979–2014 are based on Butler et al. (2017). Two additional SSW events,

on 12 February 2018 and 2 January 2019, are detected according to a wind reversal at 10 hPa and 60°N.

The MCAO response to SSW events is here defined as the change in MCAO frequency or index over a period of 30 days after the onset of the SSW (i.e., the SSW central date). The MCAO frequency ($P_M$) is defined as the percentage of days with MCAOs that exceed a 4 K threshold ($M \geq 4$ K) within a period of 30 days. The climatological MCAO frequency ($P_{M\geq4K,\text{Clim}}$) is computed between the December to March and standardized by the number of days in DJFM. Following SSW events, the

MCAO frequency ($P_{M\geq4K,\text{SSW}}$) is computed as the percentage of days with $M \geq 4$ K within a 30-day period after the SSW central date. Thus, to obtain the MCAO frequency anomaly, the MCAO frequency after SSW events is then compared with the climatological MCAO frequency.

## 3    Results

### 3.1    MCAOs in climatology and in response to SSW events

Fig. 1a demonstrates the regional distribution of the 90th percentile of the climatological MCAO index in the North Atlantic, averaged from December to March. The MCAO index is strongest over three main regions: the Labrador Sea, the Norwegian Sea and the Barents Sea (black boxes from west to east in Fig. 1a). MCAOs are also most frequent over these regions (Fig. 1b),

with a likelihood of more than 40% for an occurrence of moderate-to-strong MCAOs in the Labrador Sea during DJFM, and nearly 35% in the Norwegian Sea and the Barents Sea.

We examine the changes in the tropospheric large-scale flow conditions in response to major SSW events in ERA-Interim. Major SSW events tend to be followed by anomalously cold temperatures over the northeastern North Atlantic and Eurasia (Fig. 1c,d). These anomalies are accompanied by a north–south dipole pattern of 500-hPa geopotential height over the North Atlantic (Fig. 1e), consisting of a positive anomaly over Greenland, and a negative anomaly southeast of Greenland and over central Europe. This pattern is often associated with a negative phase of the NAO (e.g., Limpasuvan et al., 2004; Butler et al., 2017; Domeisen, 2019). Following SSW events, MCAO frequency exhibits significant regional variability, with the largest increase of MCAO frequency over the western Barents and Norwegian Seas and a decrease along the sea ice edge over the Greenland Sea as well as the Labrador Sea (Fig. 1f).

To assess whether the anomalously cold conditions, which often occur over the western North Atlantic after SSW events, have an impact on the MCAOs in these regions, we analyze changes in the MCAO frequency and strength over a period of 30 days after the onset of a SSW. We focus on the Barents Sea (70°N to 78.5°N, 30°E to 50°E, easternmost box in Fig. 1a), the Norwegian Sea (60°N to 80°N, 15°W to 5°E, central box in Fig. 1a), and the Labrador Sea (55°N to 67.5°N, 40°W to 62.5°W, westernmost box in Fig. 1a). In the next subsections, the link between the large-scale atmospheric circulation and MCAOs in these sub-regions of the North Atlantic is investigated, both in climatology and in connection with stratospheric forcing.

### 3.2 The large-scale atmospheric circulation during MCAOs

In this section we first characterize and establish the large-scale atmospheric circulation patterns associated with anomalously high MCAO occurrences in the North Atlantic for the winter months, without considering the occurrence of SSW events. For this purpose, we examine composites of the geopotential height and meridional wind anomalies in periods of moderate-to-strong MCAOs in DJFM climatology. These periods are identified using the criterion of $M \geq 4$ K. We focus on the three regions of interest shown in Fig. 1a: the Barents Sea, the Norwegian Sea and the Labrador Sea.

Periods of moderate-to-strong MCAO intensity in the Barents Sea (left column in Fig. 2) are found to be associated with a zonal dipole pattern of geopotential height anomaly in the mid-troposphere, consisting of a high-pressure anomaly over southern Greenland ("Greenland Blocking") and a low-pressure anomaly over Northern Europe, Scandinavia and the Barents Sea ("Scandinavian Trough") (Fig. 2d). Cyclone frequency during these periods indicates an increase in storminess primarily east of the Barents Sea (Fig. 2g) relative to the DJFM climatology (shown by the black contours), and a decrease in storminess over Greenland and the Irminger Sea. Consistent with cyclone frequency anomalies, surface meridional wind exhibits a negative anomaly north of Scandinavia and the Norwegian and the Barents Sea in particular, indicating an anomalous northerly flow, favourable for the advection of cold Arctic air masses over the open ocean (Fig. 2j).These atmospheric conditions are in agreement with previous studies that have shown that MCAOs in the Arctic are closely linked to the advection of cold air masses over open ocean in the cold sectors of cyclones (e.g., Kolstad et al., 2009; Jahnke-Bornemann and Brümmer, 2009; Fletcher et al., 2016; Papritz and Grams, 2018).

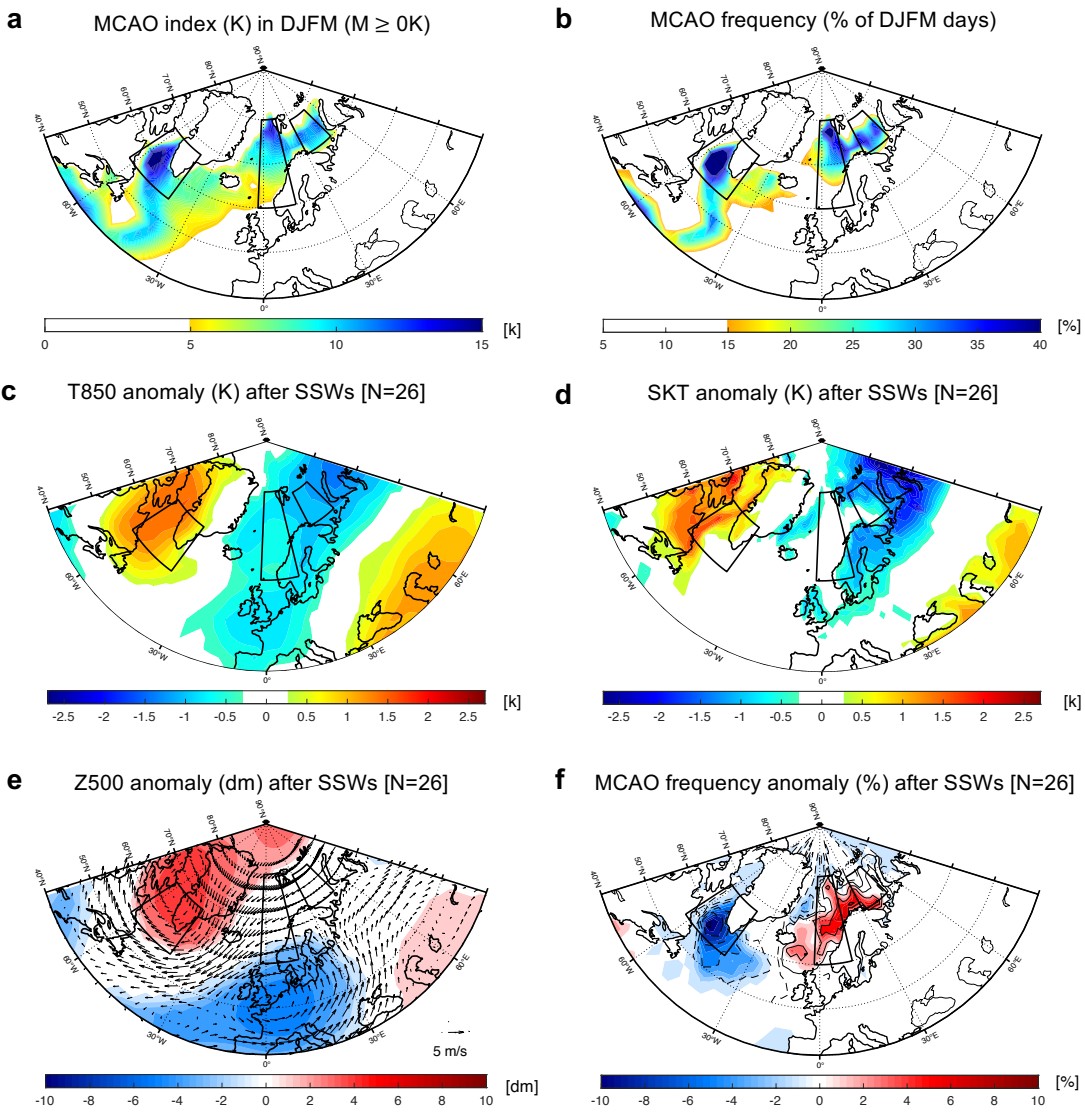

**Figure 1.** (a) The 90th percentile of the MCAO index (color shading, in K) and (b) the MCAO frequency (shading, indicated by the percentage of days for which the MCAO index exceeds 4 K) in the DJFM climatology based on ERA-Interim reanalysis. The average anomalies of (c) 850-hPa temperature (shading, in K), (d) skin temperature (shading, in K), (e) 500-hPa geopotential height (shading, in dm) and wind anomalies at the same vertical level (arrows, in m s$^{-1}$), and (f) the MCAO frequency (shading, in percentage of days out of a 30-day period. Solid and dashed contours indicate positive and negative MCAO frequency anomalies, contour interval is 2 K) over a period of 30 days following 26 observed SSW events. Black boxes show the three regions of interest: the Labrador Sea, the Norwegian Sea and the Barents Sea, where the climatological MCAO index is the highest. In all panels, only statistically significant values above the 95% level are shown by shading (based on a two-sided Student's *t-test*).

Periods of moderate-to-strong MCAOs in the Norwegian Sea (middle column in Fig. 2) are similar to those in the Barents Sea but with circulation anomalies shifted slightly to the west. Specifically, they are characterized by a zonal dipole of geopotential height anomalies, with positive anomalies centered over the region south of Greenland and negative anomalies over Scandinavia (Fig. 2e). Anomalous cyclone frequency is found to be reduced over the western Norwegian Sea, but increased along the Norwegian coast and across the Barents Sea (Fig. 2h). Consistent with that, the wind anomaly indicates a northerly flow over the Norwegian Sea (Fig. 2k). The maximum of the meridional wind anomalies is found further westward as compared to the position of the maximum meridional winds for periods of strong Barents Sea MCAOs (Fig. 2j).

During periods of moderate-to-strong MCAOs over the Labrador Sea and southern Greenland (right column in Fig. 2), finally, the geopotential height pattern is found to be associated with negative geopotential height anomalies centred over the Labrador Sea and positive anomalies over western Europe and Scandinavia (Fig. 2f). Enhanced cyclone frequency south-east of Greenland suggests a connection to increased transient storm activity over this region (Fig. 2i), associated with anomalous northerlies west of Greenland and more southerly winds south-east of Greenland and in the Norwegian Sea (Fig. 2l). Thus, this pattern is, to some extent, opposed to that found for the Barents and the Norwegian Seas but it is consistent with a southward advection of cold air masses into the Labrador Sea, as well as the subsequent eastward advection of the cold air masses south of Greenland (not shown).

Thus, more intense MCAOs in all three considered regions of the North Atlantic are clearly linked to specific large-scale circulation patterns with pronounced mid-tropospheric geopotential height anomalies over Greenland and Scandinavia, as well as related shifts of the North Atlantic storm track. These flow anomalies, in turn, cause anomalous advection of cold air masses over the open ocean west of positive cyclone frequency anomalies. In the next subsection, we investigate how the occurrence of SSW events may affect MCAOs by modulation of the prevailing tropospheric conditions. For that purpose, we will focus on the mid-tropospheric geopotential height anomalies. Establishing the link between these stratospheric events and MCAOs can provide further insight on the different pathways of MCAO occurrence, their intensity and frequency.

## 3.3 Stratospheric influence on MCAO occurrence following SSW events

In winter, extreme states of the stratospheric polar vortex can have a significant impact on the tropospheric circulation in the North Atlantic, particularly affecting the state of the NAO (e.g., Baldwin and Dunkerton, 2001; Karpechko et al., 2017; Charlton-Perez et al., 2018; Domeisen, 2019). Here, we explore the influence of the stratosphere on the large-scale tropospheric circulation in the North Atlantic, and on the intensity of MCAOs. For this purpose, we first establish the link between the large-scale circulation and MCAOs in climatology, and compare to periods that follow SSW events.

To evaluate the link between the dominant large-scale anomaly pattern and MCAOs, we first consider the extent to which the ZDI index, defined in Eq. 2, is linked to MCAO intensity. By definition, the ZDI is designed to capture the centers of action of the geopotential height anomalies.

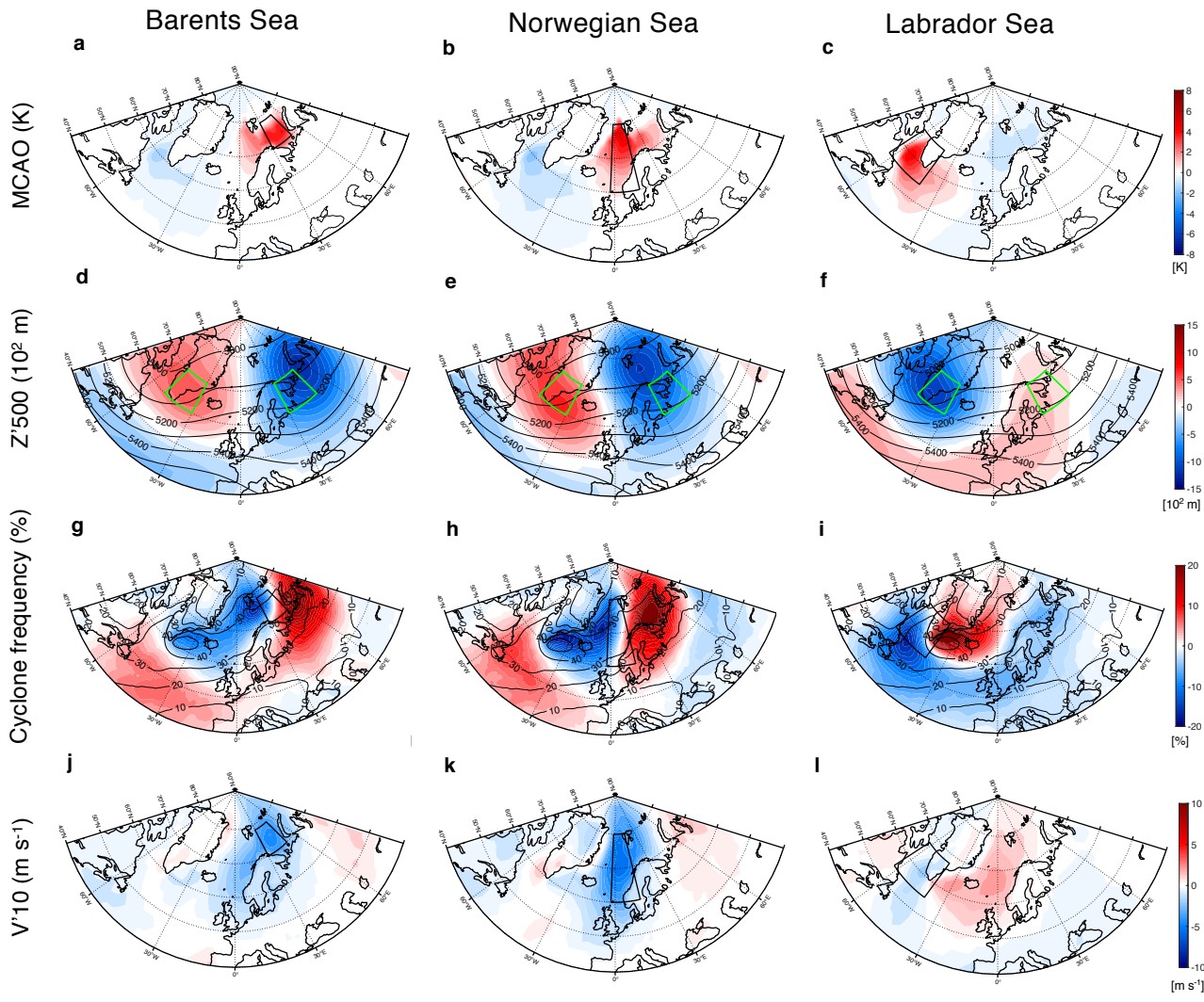

**Figure 2.** Regional flow patterns during periods of moderate-to-strong MCAO index (M ≥ 4 K) during DJFM over (left column) the Barents Sea, (middle) the Norwegian Sea, and (right) the Labrador Sea regions. Composites of daily averages of (a-c) MCAO index anomaly (color shading, in K), (d-f) geopotential height anomaly at 500 hPa (shading, in dm), and mean 500-hPa geopotential height (black contours, in intervals of 100 m) for DJFM in climatology. (g-i) Cyclone frequency anomaly (shading). The winter (DJFM) climatology of cyclone frequency is shown by black contours (starting at 10%, in intervals of 10%). (j-l) The 10-m meridional wind anomaly (shading, in m s$^{-1}$). Black boxes show the relevant region of interest: the Barents Sea, the Norwegian Sea and the Labrador Sea. Green boxes in panels d-f show the areas used in the calculation of the zonal dipole index (see section 2.3). Only statistically significant values above the 95% level are shown by the shading (based on a two-sided Student's *t-test)*.

The dependence between these indices is shown by a linear regression, as follows

$$M = b_M \cdot ZDI, \tag{3}$$

where M is the MCAO index and $b_M$ is the linear regression coefficient. In the Barents and the Norwegian Seas, a positive

linear relation is found between the ZDI index (representing the anomalous dipole pattern) and the strength of MCAOs in DJFM (black line in Fig. 3a,b). The opposite relation is found for MCAOs in the Labrador Sea region, which exhibit a negative linear relation with the ZDI index (Fig. 3c). This relation is consistent with the geopotential height pattern shown in Fig. 2f, and corresponds to a negative ZDI index.

After SSW events (triangles in Fig. 3a), a higher correlation between MCAOs in the Barents Sea and the ZDI index is found

as compared to climatology ($R^2$=0.42 after SSW events versus $R^2$=0.24 in climatology). For each SSW, this period is defined as the first 4 weeks after the onset of the SSW. Moreover, in periods that do not include SSW events, the correlation between the MCAO index and the ZDI index is weaker than in the climatology ($R^2$=0.21, shown in grey). These periods exclude the first 4 weeks after SSW events (see section 3.3). In the Norwegian Sea the correlation between the ZDI and the MCAO index increases slightly after SSW events ($R^2$=0.31) as compared to climatology ($R^2$=0.26) and periods without SSW events

($R^2$=0.24) (Fig. 3b). In the Labrador Sea, the negative correlation between the ZDI and the MCAO index is weakened after SSW events ($R^2$=0.21) relative to climatology ($R^2$=0.27). For the Barents Sea region, the correlations between the ZDI and MCAO indices for SSW (orange) and no SSW (grey) periods are significantly different from each other at the 95% confidence level using the Fisher z-test (p=0.03). However, for the Norwegian and the Labrador Sea regions the confidence level is below 95% (p=0.24 and p=0.25, respectively). The autocorrelation of the weekly indices has been accounted for by estimating the

number of independent samples, which is found to be one week in DJFM. Using a larger effective sample size after SSW events leads to a qualitatively similar conclusion.

To further understand which of the components of the dipole index has a dominant effect on the anomalous MCAO index, we separately analyse regression of the MCAO index on the geopotential height anomalies over Greenland and northern Europe as represented by $Z'_G$ and $Z'_E$ in Eq. 2, respectively. As positive anomalies of the dipole index are centered over Greenland and

negative anomalies over Scandinavia, we define the indices GB ("Greenland blocking") and ST ("Scandinavian trough") which correspond to $Z'_G$ and $Z'_E$, respectively.

The results for moderate-to-strong MCAOs following the SSW events indicate a positive correlation between the Barents Sea MCAO index and the GB index (Fig. 4a), however with a larger spread compared to the ZDI index. A negative correlation is found with the ST index, which accounts for approximately 44% of the variance after these SSWs (Fig. 4b) as compared to

the variance of 21% in climatology.

In the Norwegian Sea, a comparably high correlation is found after SSW events between the ST index and Norwegian Sea MCAOs ($R^2$=0.30) (Fig. 4d), whereas in the Labrador Sea region, a considerable correlation is found with the GB index ($R^2$=0.40) (Fig. 4e). The occurrence of SSW events somewhat weakens this correlation in the Labrador Sea compared to the periods without SSW events, although the correlation is relatively high in both cases ($R^2$=0.40 and $R^2$=0.42, respectively).

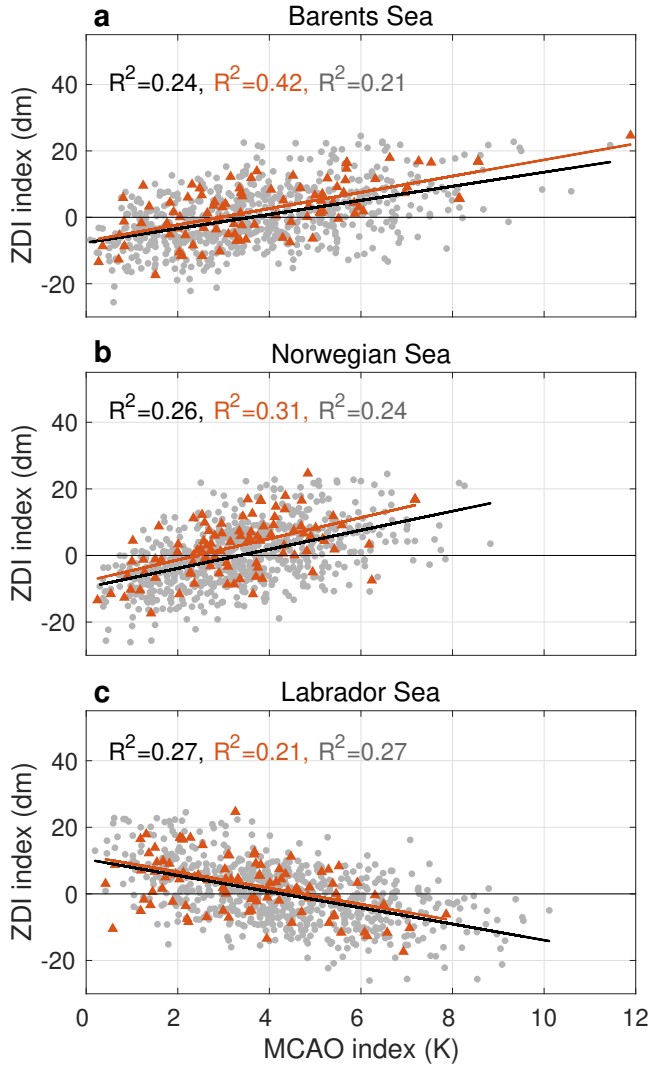

**Figure 3.** The relation between the weekly averaged ZDI index (dm) and the weekly MCAO index (K) over (a) the Barents Sea, (b) the Norwegian Sea and (c) the Labrador Sea during DJFM in ERA-Interim. Weekly averages within a 30-day period after SSW events are shown as red triangles, and weekly averages in DJFM between 1979–2019 (climatology) are marked by grey circles. Linear regression fit with corresponding $R^2$ coefficient are computed for the climatology (black) and periods following SSW events (orange). For completeness, $R^2$ for weekly averages that do not include periods after SSW events is shown in grey. All anomalies are computed with respect to the daily climatology. All correlations are statistically significant ($p < 0.05$).

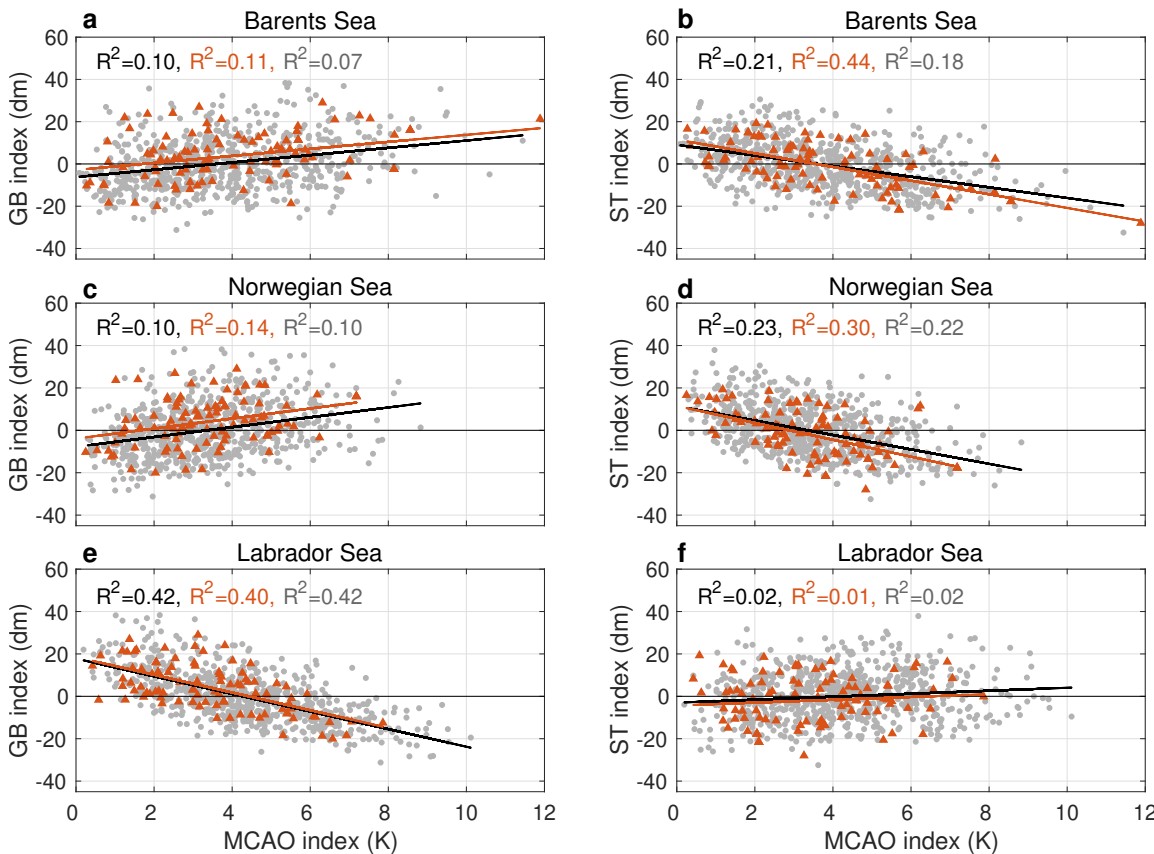

**Figure 4.** Same as Fig. 3, but for geopotential height anomalies (dm) averaged over (left) GB and (right) ST regions (green boxes in Fig. 2d, corresponding to the $Z'_{GB}$ and $Z'_{ST}$, respectively) for (a-b) the Barents Sea, (c-d) Norwegian Sea, and (e-f) Labrador Sea regions. All correlations are statistically significant ($p < 0.05$).

Out of the above correlations, only in the Barents Sea region the correlation between the ST index and MCAOs is found to be significantly different at the 95% confidence level between SSW (orange) and no SSW (grey) periods using the Fisher z-test (Fig. 4b).

     Analyzing the linear relationship between MCAOs and the ZDI index over SSW and non-SSW periods demonstrates a link between the large-scale geopotential height anomaly pattern with the strength of MCAOs in these regions. After SSW events,
there is an increase of 18% in the explained variance for the Barents Sea, and of 5% for the Norwegian Sea. The presence of a low pressure anomaly over northern Eurasia (as represented by the negative ST index) dominates the relationship in the Nordic Seas, whereas a high pressure anomaly over Greenland (a positive GB index) has a stronger relationship with the Labrador Sea MCAOs.

     Fig. 5 shows the distribution of weekly mean of ZDI, $Z'_{GB}$ and $Z'_{ST}$ indices for DJFM in periods that do not include the
first 4 weeks after SSW events (in blue), and in periods after SSW events (in orange). After SSW events, the ZDI distribution

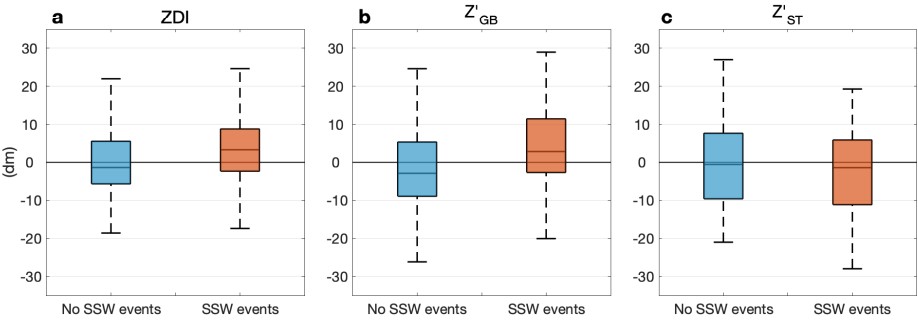

**Figure 5.** Distribution of weekly means of the (a) ZDI, (b) $Z'_{GB}$, and (c) $Z'_{ST}$ indices (in dm) for periods in DJFM that do not include the 4-weeks after SSW events (blue) and after SSW events (orange). On each box, the central line shows the median, and the bottom and top edges indicate the 25th and 75th percentiles, respectively.

generally shifts towards positive values. However, there is an overlap of the ZDI during periods with and without SSW events. The $Z'_{GB}$ index tends toward more positive values for the periods after SSW events, whereas the $Z'_{ST}$ index is rather more negative than neutral after SSW events.

Thus, a modulation of the large-scale flow patterns during DJFM and their projection on the GB and ST indices has an effect 225 on the intensity of the MCAOs in the Arctic. In particular, increasing and decreasing pressure anomalies in the centers of the large-scale zonal dipole pattern, leads to an enhancement of MCAOs in Barents and the Norwegian Seas. As stratospheric precursors such as SSWs often modulate surface weather in the European-North Atlantic regions (Charlton-Perez et al., 2018; Domeisen et al., 2020; Beerli and Grams, 2019), their impact on the large-scale circulation pattern (Fig. 1e) contributes to the increased likelihood of MCAOs in these regions in periods following SSW events (Fig. 1f).

**3.4 Relative importance of Greenland Blocking and Scandinavian Trough weather patterns for MCAOs**

In the previous section we have shown how the stratospheric influence following SSW events can modulate the dominant large-scale circulation in the European-North Atlantic region, affecting the strength of MCAOs. In winter, the Euro-Atlantic sector may be dominated by cyclonic and blocked large-scale flow features (e.g., Beerli and Grams, 2019; Papritz and Grams, 2018; Domeisen et al., 2020). In a cyclonic flow pattern, a negative geopotential height anomaly (relative to DJFM climatology) 235 dominates at 500 hPa. These negative anomalies are associated with enhanced cyclonic activity, and correspond to more than one dominant weather regime, such as Atlantic or Scandinavian Troughs (Beerli and Grams, 2019). Positive geopotential height anomalies, on the other hand, are linked to blocked weather regimes, such as Greenland blocking, and are generally associated with cold surface weather over Europe. It however remains unclear how often a strong projection of dominant geopotential height anomalies on the MCAO index occurs in climatology, and how it relates to the strength of MCAOs in the sub-regions 240 of the North Atlantic.

**Table 1.** List of observed SSW events in ERA-Interim reanalysis and their corresponding ZDI, $Z'_{GB}$ and $Z'_{ST}$ indices (units: dm), averaged over a period of 30 days following the SSW central date.

| SSW event date | ZDI | $Z'_{GB}$ | $Z'_{ST}$ |
|---|---|---|---|
| 22-Feb-79 | 0.31 | -0.02 | -0.64 |
| 29-Feb-80 | -0.62 | 5.95 | 7.20 |
| 4-Mar-81 | 3.50 | 0.02 | -6.98 |
| 4-Dec-81 | 6.83 | 12.00 | -2.03 |
| 24-Feb-84 | -1.84 | 4.33 | 8.03 |
| 1-Jan-85 | 9.91 | 12.26 | -7.56 |
| 23-Jan-87 | 11.68 | 11.00 | -12.36 |
| 8-Dec-87 | 5.32 | 0.44 | -10.19 |
| 14-Mar-88 | -1.73 | 0.25 | 3.71 |
| 21-Feb-89 | -9.61 | -14.46 | 4.75 |
| 15-Dec-98 | 1.59 | -2.07 | -5.25 |
| 26-Feb-99 | -0.48 | 7.61 | 8.57 |
| 20-Mar-00 | 5.07 | 9.66 | -0.50 |
| 11-Feb-01 | 7.87 | 6.32 | -9.43 |
| 30-Dec-01 | 6.51 | 2.53 | -10.49 |
| 18-Jan-03 | -2.60 | -2.51 | -2.67 |
| 5-Jan-04 | 0.43 | 6.06 | 5.19 |
| 21-Jan-06 | 1.96 | 6.56 | 2.63 |
| 24-Feb-07 | -6.37 | -5.08 | 7.66 |
| 22-Feb-08 | 5.58 | 0.04 | -11.12 |
| 24-Jan-09 | -1.84 | 5.04 | 8.73 |
| 9-Feb-10 | 10.62 | 15.20 | -6.03 |
| 24-Mar-10 | 1.15 | 8.83 | 6.51 |
| 6-Jan-13 | 2.23 | 0.27 | -4.19 |
| 12-Feb-18 | 1.04 | 3.65 | 1.56 |
| 2-Jan-19 | 5.88 | 5.21 | -6.56 |

As discussed in subsection 3.3, the ZDI index is found to be positively correlated to the MCAO index in the Barents and the Norwegian Seas (Fig. 3a,b). In fact, a positive ZDI occurs nearly 50% of the time in DJFM, indicating the likelihood of a dipole pattern occurrence, while an opposite dipole pattern is likely to occur for a combination of different circulation patterns. An analysis of the relation to the GB and ST geopotential height anomalies reveals that most of the variance found for the ZDI index (Fig. 3a) can be attributed to the ST index (Fig. 4a), while the relation to the GB index exhibits a much larger variability (Fig. 4b).

To assess the contribution of the zonal dipole pattern to the MCAO index in the North Atlantic, periods of GB and ST geopotential height anomalies are analyzed separately (Fig. 6). These periods are defined as days for which the 500-hPa geopotential height anomaly averaged over the GB and ST boxes is positive or negative, respectively. Results show that for both positive GB (Fig. 6a) and negative ST (Fig. 6b) the circulation over the Barents Sea is anomalously cyclonic (Fig. 6c,d), and associated with an increase in the MCAO index over the Barents and the Norwegian Seas (Fig. 6e,f). Interestingly, only the GB pattern is accompanied by a reduced frequency of MCAOs in the Labrador Sea. These differences in MCAO are clearly related to the differences in storminess; Periods of negative ST are associated with increased storminess over Scandinavia and the southern Barents Sea, whereas periods of GB exhibit a strong reduction in cyclone frequency over the Nordic Seas centered over the Irminger Sea.

Furthermore, we examine the dependency of the MCAO index on the GB and ST indices in these sub-regions of the North Atlantic during DJFM. In the Barents Sea, stronger MCAOs (represented by light blue marker color) are found to be associated with negative ST index, for either a positive or a negative GB index (Fig. 7a). The most intense MCAOs are found for a negative ST index and a positive GB index, emphasizing the importance of this particular combination for the occurrence of MCAOs in the Barents Sea (this pattern is consistent with a positive ZDI index). In the Norwegian Sea, the MCAO dependency on the GB and the ST indices is found to be similar to that of the Barents Sea, with stronger MCAOs associated with a negative ST index (Fig. 7b). In contrast, in the Labrador Sea, stronger MCAOs are primarily associated with a negative GB index, demonstrating a weaker sensitivity to the sign of the ST index (Fig. 7c).

## 4    Conclusions

This study focuses on the influence of the stratosphere on the occurrence of marine cold air outbreaks in the North Atlantic and their connection to the large-scale circulation patterns over the North Atlantic and Europe. Particularly, we investigate how the frequency and the magnitude of such MCAOs in the Barents Sea, the Norwegian Sea, and the Labrador Sea are affected by the large-scale conditions after the onset of extreme events in the stratosphere, known as SSW events.

By analyzing the regional atmospheric conditions in DJFM between 1979 to 2019 we find that a positive 500-hPa geopotential height anomaly over Greenland and a negative geopotential height anomaly over Scandinavia, accompanied by increased storminess and northerly surface winds over the Barents Sea and to the east of the Barents Sea, are strong indicators for enhanced MCAO intensity in these regions. In contrast, the opposite geopotential height anomaly pattern (i.e., lower geopotential height anomaly over Greenland and higher geopotential height anomaly over Scandinavia) and increased storminess in the

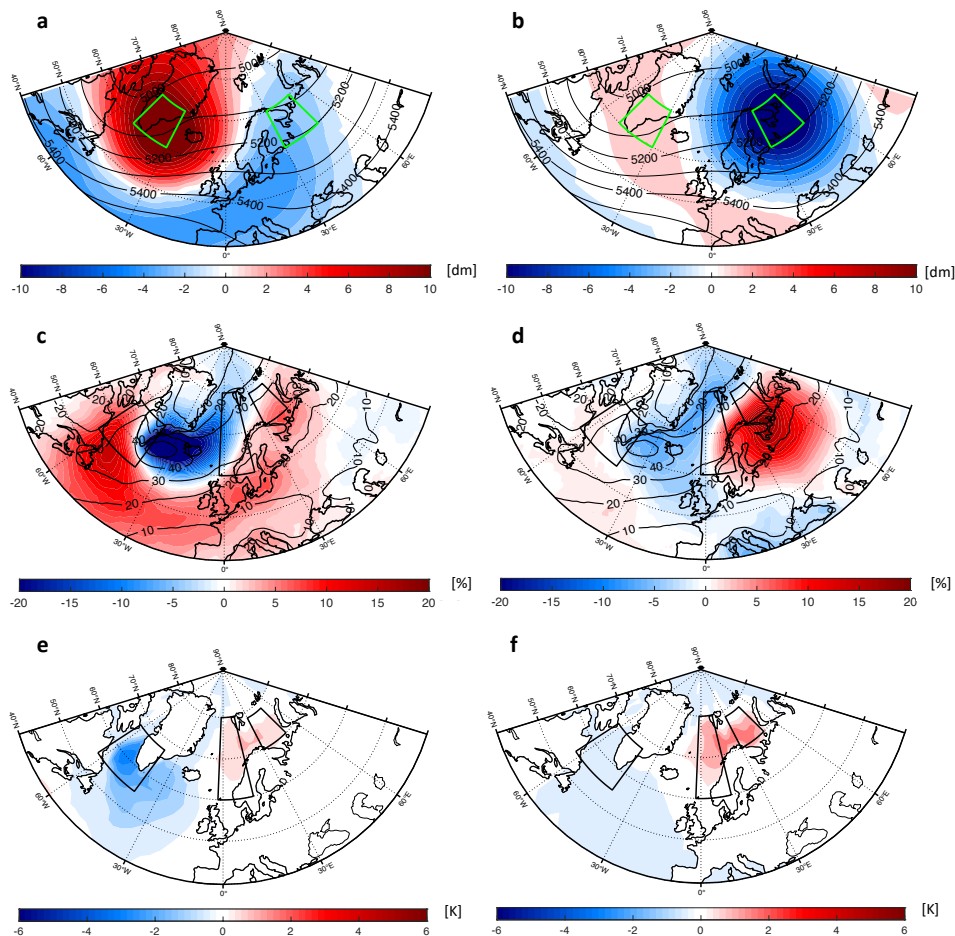

**Figure 6.** Regional patterns for the Greenland Blocking (GB) and Scandinavian Trough (ST) composites in DJFM. (a,b) Geopotential height anomaly at 500 hPa (color shading, in dm) and mean 500-hPa geopotential height (black contours, in intervals of 100 m) for DJFM in climatology, (c,d) mean cyclone frequency (shading, in %), and (e,f) MCAO index anomaly (color shading, in K) for days with (left column) positive 500-hPa geopotential height anomaly averaged over the GB box, and (right column) negative anomaly averaged over the ST box, respectively. Black contours in panels (c,d) show the DJFM climatology of cyclone frequency (from 10%, in intervals of 10%). Only statistically significant values above the 95% level are shown (based on a two-sided Student's *t-test).

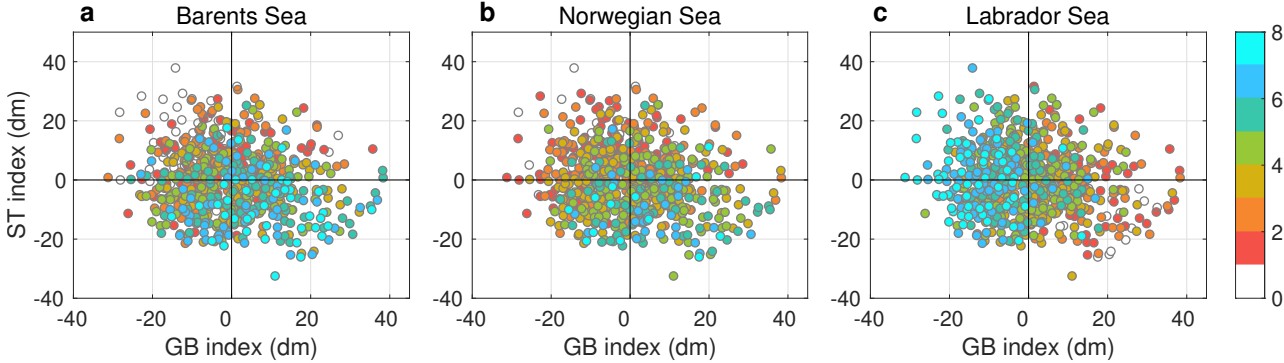

**Figure 7.** Scatter plot of weekly means of 500-hPa geopotential height anomalies (in dm) averaged over the GB and ST regions (westernmost and easternmost green boxes in Fig. 2d, respectively) during DJFM for (a) the Barents Sea, (b) the Norwegian Sea, and (c) Labrador Sea regions. As in Fig. 3, all anomalies are computed with respect to the daily climatology and averaged weekly. Marker color indicates the mean MCAO index (K) averaged over each of the basins (separately for each panel).

Irminger Sea are found to be associated with stronger MCAOs in the Labrador Sea. These circulation patterns highlight the connection between MCAOs in the Arctic and the cold sectors of cyclones, in agreement with previous studies (e.g., Fletcher et al., 2016; Papritz and Grams, 2018).

After SSW events, significant changes in the tropospheric surface flow tend to occur. These changes involve a negative phase of the NAO and extremely cold temperatures over the northeastern North Atlantic and Northern Europe (Fig. 1c,d). To assess whether these extreme changes have an impact on the occurrence of MCAOs in these regions, we analyze the characteristics of the large-scale atmospheric conditions after 26 SSW events between 1979 to 2019, as compared to climatology. We find that changes in the large-scale atmospheric circulation pattern, represented by a positive zonal dipole index, accounts for 42% of the MCAO variance in the Barents Sea and 31% of the variance in the Norwegian Sea after SSW events. For comparison, the dependency on the zonal dipole index explains only 21% and 24% of the variance in winters without SSW events, respectively. Thus, the correlation between the zonal dipole index and MCAOs following SSW events is found to be significantly higher than the correlation between the zonal dipole index and MCAOs in periods without SSWs (section 3.3).

Furthermore, we find that an intensification of the positive geopotential height anomaly over the southern part of Greenland (as represented by a positive Greenland Blocking pattern) is associated with weaker MCAOs in the Labrador Sea, and accounts for ∼40% of the MCAO variance in this region, both in periods following SSW events and in the DJFM climatology in general. While the relevance of stratospheric forcing for MCAOs in the Labrador Sea is not found to be as statistically significant as in the Barents and the Norwegian Seas, the importance of MCAOs for dense water formation in this region implies that SSWs might nevertheless have an impact on the North Atlantic overturning circulation, as suggested by Reichler et al. (2012).

Through linear regression analysis we demonstrate a statistical relationship between MCAOs and atmospheric indices that capture the characteristics of the large-scale flow (Figs. 3, 4). Such a connection can be further used for mitigation of societal and economic impacts by providing an estimate of the increase/decrease in MCAO intensity due to a change in the environ-

mental conditions. Furthermore, understanding the connection between MCAOs in the North Atlantic and the stratospheric forcing shows potential for improved predictive skill of cold air outbreaks on subseasonal to seasonal time scales. Recently, Polkova et al. (2019) analyzed prediction skill for MCAOs over the Barents Sea using the seasonal prediction system based on the Earth System Model from the Max-Planck Institute for Meteorology. According to their analysis, MCAOs can be predicted at lead times of about 2.5 weeks, starting from November initial conditions. Our results show that stratospheric precursors on subseasonal timescales lead to an increased likelihood of the favorable conditions for MCAOs in the Barents and Norwegian Seas. Thus, this connection can potentially be exploited for improving subseasonal MCAO predictions.

The preferred patterns for MCAOs may also indicate the pathway of cold air masses during MCAO formation. MCAO air masses over the Barents Sea tend to originate in the high or Siberian Arctic, with dominant pathways of cold air masses from Siberia across Novaja Zemlja and the northern sea ice edge into the Barents Sea (Papritz and Spengler, 2017). We show that the northern pathway is largely captured by a positive ZDI (Fig. 2d) and is consistent with a low pressure anomaly over north-eastern Europe, bringing cold air masses southward across the Norwegian and the Barents Seas (Fig. 2j,k). A negative ZDI, on the other hand, can be linked to a dominant blocking pattern over the Barents and Kara Seas, possibly related to the eastern pathway for MCAOs in the Barents Sea, consistent with Papritz (2017). A similar link between a positive ZDI index and MCAOs is found over the Norwegian Sea, suggesting the relevance of a northern pathway for the development of Norwegian Sea MCAOs (Fig. 2h,k). In contrast, a pathway for MCAOs in the Labrador Sea is linked to a dominant cyclonic regime over Greenland, bringing a flow of cold air southward into the Labrador Sea (Fig. 2i,l).

We conclude that understanding the connection between the stratosphere and the occurrence of MCAOs in the North Atlantic reveals key ingredients for MCAO formation, which can potentially lead to improved prediction skill on subseasonal time scales due to the long-lasting circulation anomalies associated with stratosphere-troposphere coupling in winter. SSW events are found to have an effect on the large-scale circulation pattern in the troposphere, as evident from the ZDI distribution shift towards positive values after SSW events (Fig. 5a). There is, however, a large variability among SSW events, as also discussed in previous studies (e.g., Karpechko et al., 2017; Afargan-Gerstman and Domeisen, 2020; Domeisen et al., 2020), which imposes some limitations on the predictability that in principle can be obtained in terms of MCAO forecasting.

In addition to the influence of the large-scale tropospheric flow, conditions in the boundary layer also play a role in the formation of MCAOs in the Arctic. Northerly winds during strong MCAOs in the Barents, Norwegian and Labrador Seas are found to be stronger than climatology (Fig. 2j-l), consistent with previous studies (Kolstad, 2017; Fletcher et al., 2016). Another factor that affects the occurrence of MCAOs is sea ice cover in the Barents Sea (e.g., Ruggieri et al., 2016). In a warming climate, the diminishing sea ice cover over the Barents Sea can potentially modulate MCAO occurrence in this region, by exposing more of the ocean surface to interaction with the atmosphere. Such changes are also likely to be affected by the availability of cold air masses in the Arctic (e.g., Papritz et al., 2019). Further work is thus required for understanding a compound effect of Arctic processes on MCAO formation.

*Data availability.* ERA-Interim reanalysis has been obtained from the ECMWF server (https://apps.ecmwf.int/datasets/data/interim-full-daily, last access: July 2020) (Dee et al., 2011).

*Author contributions.* H.A-G. and D.D. initiated and led the study. I.P., J.B., P.R., and M.K. contributed ideas on the early stage of the study. H.A-G. performed the analysis and created all figures and tables. L.P. provided the cyclone frequency datasets and contributed to the interpretation of the results. H.A-G. wrote the manuscript with contributions from all authors.

*Competing interests.* The authors declare no competing interests.

*Acknowledgements.* Funding to H.A-G., I.P., P.R., M.K., P.A., J.B. and D.D. was provided by the Blue-Action project from the European Union's Horizon 2020 research and innovation programme under grant agreement No 727852. ERA-Interim reanalysis has been obtained

from the ECMWF server (https://apps.ecmwf.int/datasets/data/). Funding to D.D. and H.A-G. by the Swiss National Science Foundation through grant No. PP00P2_170523 is gratefully acknowledged.

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
