# Peer review of "Stratospheric influence on North Atlantic marine cold air outbreaks following sudden stratospheric warming events"

_Weather and Climate Dynamics, 2020_

## Referee Comment (RC1) · Anonymous Referee #2 · 26 May 2020

The authors study stratospheric influence on Marine Cold Air Outbreaks (MCAO) in the Barents Sea using reanalysis data. They show that Sudden Stratospheric Warmings are followed by enhanced frequency of occurrence of MCAO in the Barents Sea. They suggest that this connection can potentially lead to improved predictability of MCAO on sub-seasonal to seasonal time scales. I believe the paper explores an interesting topic which has potentially interesting implications for forecasting MCAO at longer lead times. However, the paper fails, in my opinion, to thoroughly document stratospheric influence on MCAO and to convincingly demonstrate that the stratosphere plays a role in occurrence of MCAO in the Barents Sea. There are several points that the authors demonstrate rather clearly: (1) Climatologically, MCAO occur most frequently over Barents, Norwegian and Labrador Seas (Fig. 1a,b). (2) MCAO are more frequent over Norwegian and Barents seas and less frequent over Labrador Sea following SSWs (Fig. 1d). (3) High over Greenland and low over Scandinavia favour MCAO occurrence in the Barents Sea (Fig. 5). (4) Anomalies over both regions contribute to MCAO occurrence (Fig. 6). I believe these points are clear, but I suspect they may be not particularly new. However, the rest of the manuscript concerning the stratospheric role in MCAO occurrence is less clear to me. Specific issues, which need to be addressed before possible publication, are listed below. I recommend major revision and encourage the author to revise the manuscript and resubmit it.

Major problems:

(1) Selection of the region: Why the authors choose to focus on the Barents Sea region while it seems from Fig. 1d that SSW impacts in the selected easternmost box is nearly absent? The selection is more puzzling since the authors state in the introduction that the interest to MCAO is triggered by the risks they pose on populated Norwegian coast. The Norwegian sea region, which has enhanced frequency of MCAO occurrence following SSWs, seems to be a more logical choice.

(2) Sub-selection of SSWs: SSWs have divergent surface impacts and not all SSWs have significant surface impacts, which, in general, justifies sub-selecting only interesting events. However, selecting events based on impacts over a small region located at the edge of the canonical SSW fingerprint is dangerous because of a small signal to noise ratio. But this is exactly what the authors do. Further, the criterion that the authors choose for selecting the events seems to be very relaxed. It may be that some of the events the authors selected are not necessarily represent stratosphere-troposphere coupling. For example, December 2001 event was followed by positive tropospheric NAM during most of the winter, which is inconsistent with established stratospheric influence on the troposphere. Situation in January 2003 was similar. It is likely that internal tropospheric dynamics played a more important role during these winters. Nevertheless, both these events are listed as those having stratospheric impact on Barents Sea MCAO. My question is, what if you randomly select 24 dates over

the ERA-I period and calculate MCAO frequency during 30 days following these dates? How likely is to get 8 events followed by periods with 30% of MCAO occurrence simply by chance, given that the climatological frequency of MCAO occurrence in the region is about 23%, i.e. not much less chan 30%?

(3) Mechanism of SSW influence: Table 1 shows that ZD index is positive for most SSWs, which suggests that it may play a role in MCAO occurrence following SSWs. However, the selection of SSWs is not based on ZD index. The March 1981 and February 2001 events both have strong ZD index but they are not selected. Why not include them? Going into more details, Fig. 4 shows that Scandinavian trough is more important for MCAO intensity then Greenland blocking. However, according to Table 1 SSW has a stronger signal over Greenland, while it has insignificant influence on Scandinavian trough. Here, 11 out of 24 events show positive Z anomaly, inconsistent with proposed mechanism. Further, even the selected 8 events do not have a consistent signal in either Scandinavian trough or in Greenland blocking regions. Thus, I wonder how MCAO forecasting can benefit from SSW predictability if the mechanism of influence is not well established?

(4) Abstract (L 5-6) says that "Overall, more than a half of SSW events lead to more frequent MCAOs in the Barents Sea." However this is not supported by Figure 2 which shows that exactly half of SSWs are followed by reduced frequency of MCAO in the Barents Sea.

(5) Figure 4 shows that stronger ZD index corresponds to more intense MCAO, likely through more intense northerly flow. But what is the purpose of showing the correlation separately for periods after SSWs? What is the physics behind apparently higher correlation between ZD index and MCAO intensity during periods following SSWs?

(6) I am surprised that the authors did not collect the data for the two recent SSW events. Surface impacts by SSWs have a very low signal-to noise ratio. For example, Maycock and Hitchcock (2015) showed that a large number of events (about 50) are

required to detect difference between impacts by splits and displacements. Establishing significant signal seem to be important also for your paper. Adding two recent SSW events would increase the sample size by 8% which is a considerable improvement. The dates for the 2018 and 2019 events could be found for example in the following paper: Afargan-Gerstman, H., & Domeisen, D. I. V. (2020). Pacific modulation of the North Atlanticstorm track response to suddenstratospheric warming events.Geophysical Research Letters,47,e2019GL085007. https://doi.org/10.1029/2019GL085007.

Reference: Maycock, A. C., and P. Hitchcock (2015), Do split and displacement sudden stratospheric warmings have different annular mode signatures?, Geophys. Res. Lett., 42, 10,943–10,951, doi:10.1002/2015GL066754.

Other comments:

L50 After reading the paper I was wondering whether the Barents regions discussed in the paper is so relevant for the densely populated Norwegian coast?

L44 "weak stratospheric forcing" Is it a combination of "weak stratospheric vortex" and "stratospheric forcing"?

L88-90: How do you calculate MCAO frequency and frequency change after SSW? Please provide equations.

L109-110: "Using this classification, we are able to capture the favorable conditions for MCAO occurrence in response to stratospheric forcing." Are these conditions different from those that favor MCAO occurrence without stratospheric forcing?

L191: Should Fig. 5a be replaced by Fig. 4a.

L220: "often followed by a more frequent occurrence" Strange expression. I don't think that saying "It often occurs more frequently" makes much sense but it is what the authors are trying to say.

---

## Referee Comment (RC2) · Anonymous Referee #1 · 29 May 2020

**General comments:**

In this manuscript, the authors evaluate whether there is a relationship between marine cold air outbreaks (MCAOs) and Sudden Stratospheric Warmings (SSWs) in the Barents and Norwegian Seas. The authors make the conclusion that 33% of SSWs are associated with an enhanced MCAO response in the Barents Sea. They furthermore conclude that a positive zonal dipole pattern in the large-scale atmospheric flow accounts for 44% of the MCAO variance in the Barents Sea. This manuscript fits within the scope of WCD in that it addresses stratosphere-troposphere coupling, and prediction on subseasonal to seasonal time scales. The authors present convincing evidence that MCAOs in the North Atlantic are most frequent over the Barents, Norwegian, and Labrador Seas, while MCAOs are more frequent in the Barents and Norwegian Seas

in a 30-day period following SSW events. However, I do not think this is strictly a new result (e.g., Fletcher et al. 2016). There is also a convincing case that the Zonal Dipole Index (ZDI) and MCAO are more correlated in the 30-days after an SSW. A key here though is that it is 'more' correlated, and it is not clear what threshold needs to be met in order for there to be a meaningful relationship. Furthermore, the composite patterns after SSWs (Fig. 3) and with MCAOs (Fig. 5) are only roughly similar. Overall, it is my opinion that while this manuscript has some promise, the results are far too premature for publication in WCD at this time. In particular:

1. The term 'enhanced MCAO' refers to "SSW events with an MCAO frequency response above a threshold of 30% are classified as SSWs with a strong MCAO response in the Barents Sea." This 30% is quite arbitrary and no justification is provided and is completely what determines the 33% value in their main conclusion.

2. There is not much of a physical connection with how large-scale fields at 500 hPa and 300 hPa connect to cold air at the surface. Given the frequent elevated inversions in the Arctic, it is not clear under what circumstances the boundary layer couples with the free tropospheric fields. The processes described by Pithan (2018) and Papritz et al. (2019), for example, may be good to incorporate.

3. To test the relationship between the ZDI and MCAO index, the authors perform a linear regression and show some scatterplots (Fig. 3), arguing that $R^2$ is higher just after SSW events compared to climatology. True, the values are higher ($R^2 = 0.44$ vs. $R^2 = 0.15$ in Fig. 3a), but what value of $R^2$ would have made the authors conclude that there is not a difference. $R^2 = 0.44$ would not be considered high in many circumstances, so why here?

4. There is not convincing evidence that the meridional wind at 300 hPa relates to meridional wind at the surface advecting cold air masses. It is argued that an $R^2$

of 0.13 is more meaningful than an $R^2$ of 0.12 to conclude that there is a stronger relationship between meridional wind and MCAO index in the aftermath of SSW events (line 145). Why not 850 hPa meridional wind (or lower)?

5. Table 1 in Butler et al. (2017) provides 24 historical major SSW events between 1979-2014, but the authors analyze atmospheric fields and climatologies from 1979-2016. Thus the range for the analysis period of this study can only be limited through 2014 only. Furthermore, the authors begin with a seasonal December, January, February (DJF) analysis, then switch to also include March (DJFM) midway through (Line 130). They should always use DJFM given that the SSWs contain March events.

6. Why is the climatology not following a standard 30-year climatology as recommended by the WMO (World Meteorological Organization 2017)? Usually it is 1979-2010.

**Specific comments:**

1. Table 1 in Butler et al. (2017) provides 24 historical major SSW events between 1979-2014, not 2016 as stated. This limits the range for the analysis period of this study to be through 2014 only.

2. The analysis corresponding to Figures 1 and 2 is for DJF, when some of the SSW events extend into March as the authors point out (but not until Line 130). Authors should consistently use DJFM instead of strangely adding March 'midstream' on Line 130.

3. How are the geographic boxes for the Zonal Dipole Index (ZPI) determined (shown in Fig. 3b)? Justification is needed.

4. Line 50: I assume by 'predictability', the authors actually mean 'practical predictability.' Otherwise elaboration is needed.

5. Line 75: $Z_G$ and $Z_E$ should be $Z'_G$ and $Z'_E$, respectively, given the definition of geopotential height anomaly from climatology in Line 74. Similarly in equation (2).

6. Line 90: Insert 'in the North Atlantic' after 'MCAO index'

7. Line 105: Authors state 'More than half of the SSW events' when I count 12/24 from Figure 2 that are above 25%. Also, the text states that the mean MCAO frequency in DJF is 25%, but in the figure, it looks like 24%. So if it were 25%, that would remove at least one more sample to be less than half.

8. Line 120: It is redundant to plot anomalous 500 hPa meridional wind on a 500 hPa geopotential height anomaly plot since flow could reasonably be assumed to be quasi-geostrophic. Furthermore, it is not very convincing to assume that northerly winds at 500 hPa extend to the surface where the cold air outbreak occurs. If the point of this panel is to evaluate the large-scale flow and how it may differ from the average flow during events, it would be more informative to plot the mean 500 hPa height contours instead.

9. Line 175: The 500-hPa height patterns look quite different between Figures 3b and 5b. After SSWs (Fig. 3b), it the anomalies suggest there is a breaking Rossby wave pattern consistent with LC1 (Thorncroft et al. 1993) with a cut-off low over Central Europe, which is a very different pattern than for the strong MCAOs in the Barents Sea (Fig. 5b). This suggests that on average, the patterns may be considerably different, thus limiting how this relationship could be applied in any prognostic form.

10. The text should be reserved to discuss the results of figures, and the caption should provide instructions on how to read and interpret the figure. Lines 115 and 200 are examples where the main text repeats the information in the caption and disrupts the flow of the narrative (Fig. 3b shows..., Fig. 6 shows...).

11. What are the contours in Figure 3c?

**Technical corrections:**

1. Line 65: The inline subscripts 'skt' and '850hPa' below equation (1) should be in text mode, consistent with those in equation (1).

2. Lines 75 and 135: ZDI is text, and should be written in text mode, not math mode.

3. Line 105: a half → half

4. There should be a space between numbers and units. This occurs frequently with '4K', for example on line 70.

**References:**

Butler, A. H., J. P. Sjoberg, D. J. Seidel, and K. H. Rosenlof, 2017: A sudden stratospheric warming compendium. Earth System Science Data, 9.

Fletcher, J., S. Mason, and C. Jakob, 2016: The climatology, meteorology, and boundary layer structure of marine cold air outbreaks in both hemispheres. J. Climate, 29 (6), 1999-2014.

Papritz, L., E. Rouges, F. Aemisegger, and H. Wernli, 2019: On the thermodynamic pre- conditioning of arctic air masses and the role of tropopause polar vortices for cold air outbreaks from Fram strait. J. Geophys. Res.: Atmos.

Pithan, F., et al., 2018: Role of air-mass transformations in exchange between the Arctic and mid-latitudes. Nat. Geosci., 11 (11), 805.

Thorncroft, C. D., B. J. Hoskins, and M. E. McIntyre, 1993: Two paradigms of baroclinic-wave life-cycle behaviour. Quart. J. Roy. Meteor. Soc., 119, 17-55.

World Meteorological Organization, 2017: WMO guidelines on the calculation of climate normals. World Meteorological Organization Switzerland.

[Figure]

**[WCDD](https://doi.org/10.5194/wcd-2020-11)**

---

## Referee Comment (RC3) · Anonymous Referee #3 · 30 Jun 2020

The motivation for the article is worthwhile and will be interesting to readers once a few issues are addressed. In its current form, there are a few gaps in the analysis that need to be filled before publication. I share some concerns with the other two reviewers.

- The first major issues centers on using the top tercile of events as the focus the analysis results. By subsetting of the SSWs into the top tercile of MCAO response, the readers only sees cases that fit one storyline. Fig 2 shows that this MCAO storyline is not always consistent across all SSWs. Since the authors suggest their analysis would inform decision makers that use S2S forecasts, one way to address this issue is by adding analysis of SSW events with non/weak MCAO responses for comparison. Such a solution could involve a corresponding analysis of the bottom tercile events. In such an analysis broader questions could be answered, e.g., are there mechanisms

of troposphere-stratosphere coupling that occur in the post-SSW period that favor the enhanced/suppressed MCAO response? Such an analysis would provide readers with needed context for the interpretation of the extreme MCAO response.

- Adding an analysis along these lines to the article would also get at addressing a second major issue, the connection between the stratosphere and the large-scale tropospheric flow is assumed and not shown as part of this work in its present form. Is there a stratospheric flow metric that would provide an indication of the likelihood of whether or not the 30-day period after the SSW would have a bottom or top tercile event or is it not possible to determine at the SSW onset date? With the addition of such an analysis, the authors could potentially provide insight into the type/evolution of SSW events required for the high-impact MCAO response.

──────────────────────────

---

## Author Comment (AC1) · 5 Aug 2020

The comment was uploaded in the form of a supplement:
https://wcd.copernicus.org/preprints/wcd-2020-11/wcd-2020-11-AC1-supplement.pdf

---

## Author Response (AR1)

**Response to Reviewers**

We would like to thank the reviewers for the useful comments for our study. Following the reviewers' reports, we have made several changes to the manuscript, and extended both the scope of the manuscript as well as the time series used.

In particular, we address the issues raised by the reviewers regarding the sub-selection of SSW events based on the marine cold air outbreak response in the Barents Sea. We have revised our analysis to consider all SSW events, rather than sub-selecting particular SSW events. In addition, we have extended our analysis to explore marine cold air outbreak in three regions of increased cold air outbreak frequency in the North Atlantic: the Barents Sea, the Norwegian Sea and the Labrador Sea (instead of considering the Barents Sea only). By extending the analysis to the entire North Atlantic, we are able to compare the dominant pathways for MCAOs in the Barents and Norwegian Seas to those of the Labrador Sea, and shed light on the spatial variability of the MCAO response after SSW events (shown in Fig. 1f).

Please find below a point-by-point reply (in blue) to the reviewers' comments and suggestions. All changes are highlighted in the attached version of the manuscript.

**Reviewer 1**

The authors study stratospheric influence on Marine Cold Air Outbreaks (MCAO) in the Barents Sea using reanalysis data. They show that Sudden Stratospheric Warmings are followed by enhanced frequency of occurrence of MCAO in the Barents Sea. They suggest that this connection can potentially lead to improved predictability of MCAO on sub-seasonal to seasonal time scales. I believe the paper explores an interesting topic which has potentially interesting implications for forecasting MCAO at longer lead times. However, the paper fails, in my opinion, to thoroughly document stratospheric influence on MCAO and to convincingly demonstrate that the stratosphere plays a role in occurrence of MCAO in the Barents Sea.

There are several points that the authors demonstrate rather clearly:

- (1) Climatologically, MCAO occur most frequently over Barents, Norwegian and Labrador Seas (Fig. 1a,b).

- (2) MCAO are more frequent over Norwegian and Barents seas and less frequent over Labrador Sea following SSWs (Fig. 1d).

- (3) High over Greenland and low over Scandinavia favour MCAO occurrence in the Barents Sea (Fig. 5).

- (4) Anomalies over both regions contribute to MCAO occurrence (Fig. 6).

I believe these points are clear, but I suspect they may be not particularly new. However, the rest of the manuscript concerning the stratospheric role in MCAO occurrence is less clear to me. Specific issues, which need to be addressed before possible publication, are listed below. I recommend major revision and encourage the author to revise the manuscript and resubmit it.

**Major problems:**

(1) Selection of the region: Why the authors choose to focus on the Barents Sea region while it seems from Fig. 1d that SSW impacts in the selected easternmost box is nearly absent? The selection is more puzzling since the authors state in the introduction that the interest to MCAO is triggered by the risks they pose on populated Norwegian coast. The Norwegian sea region, which has enhanced frequency of MCAO occurrence following SSWs, seems to be a more logical choice.

We thank the reviewer for this comment. Initially, the decision to focus on the Barents Sea was based on its importance for many aspects of human activity in the Arctic. In particular, there is a considerable marine activity in the Barents Sea region, mainly due to the oil and gas industry, fishing, and the growing global

interest in new shipping routes to/from Asia. Small-scale extreme weather events, such as polar-lows, can cause substantial damage to the marine activity in the Barents Sea region, in addition to the already-mentioned risks to the highly-populated Norwegian coast. Even in the absence of polar lows, intense surface winds can form along Arctic fronts (e.g., Kolstad, 2017, Pithan et al., 2018).

In the revised version, we extend the scope of the analysis to two additional regions of increased cold air outbreak frequency in the North Atlantic: the Norwegian Sea and the Labrador Sea regions. As both the Norwegian Sea (or the extended region known as Greenland–Iceland–Norwegian (GIN) Seas) and the Labrador Sea experience a high frequency of MCAOs in winter (Kolstad et al., 2009, Fletcher et al., 2016), we investigate the stratospheric influence on MCAO occurrence in these regions. All figures in the manuscript have been updated accordingly.

(2) Sub-selection of SSWs: SSWs have divergent surface impacts and not all SSWs have significant surface impacts, which, in general, justifies sub-selecting only interesting events. However, selecting events based on impacts over a small region located at the edge of the canonical SSW fingerprint is dangerous because of a small signal to noise ratio. But this is exactly what the authors do. Further, the criterion that the authors choose for selecting the events seems to be very relaxed. It may be that some of the events the authors selected are not necessarily represent stratosphere- troposphere coupling. For example, December 2001 event was followed by positive tropospheric NAM during most of the winter, which is inconsistent with established stratospheric influence on the troposphere. Situation in January 2003 was similar. It is likely that internal tropospheric dynamics played a more important role during these winters. Nevertheless, both these events are listed as those having stratospheric impact on Barents Sea MCAO. My question is, what if you randomly select 24 dates over the ERA-I period and calculate MCAO frequency during 30 days following these dates? How likely is to get 8 events followed by periods with 30% of MCAO occurrence simply by chance, given that the climatological frequency of MCAO occurrence in the region is about 23%, i.e. not much less than 30%?

We thank the reviewer for this comment. In the revised manuscript, several changes have been made to address this issue. First, we have removed the sub-selected SSW events based on their impact in the Barents Sea. Instead, we analyze all SSW events in the historical record. Thus, we do not focus on 8 out of 24 events with a particular response, and no criterion is applied. In addition, we extended the reanalysis dataset to 2019 in order to include two recent SSW events in our analysis (in total: 26 events. See Table 1 in the revised manuscript for full list of events).

During SSWs, changes in the stratospheric circulation can significantly influence surface weather in the Northern Hemisphere, through coupling between the stratospheric polar vortex and the tropospheric jet stream (Baldwin and Dunkerton, 2001; Charlton et al., 2003; Domeisen et al., 2019; Scaife et al., 2005). However, as the also mentioned by the reviewer, the tropospheric response following SSW events can exhibit a rather large variability (e.g., Afargan-Gerstman and Domeisen, 2020, Domeisen et al. 2020), which imposes some limitations on the predictability of MCAOs in the North Atlantic. Therefore, following the reviewer's comment, we have removed the sub-selection of events, and revised Fig 1 to show only statistically significant anomalies above the 95% level. In addition, to estimate the significance of the difference between correlations after SSW and non-SSW periods, we use the Fisher z-transform at the 95% significance level. Several sentences on these results have been added to the manuscript (lines 187-190 and 198-201), and all figures have been revised accordingly.

(3) Mechanism of SSW influence: Table 1 shows that ZD index is positive for most SSWs, which suggests that it may play a role in MCAO occurrence following SSWs. However, the selection of SSWs is not based on ZD index. The March 1981 and February 2001 events both have strong ZD index but they are not selected. Why not include them? Going into more details, Fig. 4 shows that Scandinavian trough is more important for MCAO intensity then Greenland blocking. However, according to Table 1 SSW has a stronger signal over Greenland, while it has insignificant influence on Scandinavian trough. Here, 11 out of 24 events show positive Z anomaly, inconsistent with proposed mechanism. Further, even the selected 8 events do not have a consistent signal in either Scandinavian trough or in Greenland blocking regions. Thus, I wonder how MCAO forecasting can benefit from SSW predictability if the mechanism of influence is not well established?

Following the reviewer's comments, we have revised our analysis to include all SSW events (without sub-selection of specific SSWs). Thus, both the SSW of 1981 and the SSW of 2001 (which the reviewer mentions) are now taken into consideration. For consistency, we have also updated the values in Table 1 to show ZDI, $Z'_{GB}$ and $Z'_{ST}$ averages for a period of 30 days after each of the SSWs. As pointed out by the reviewer, the downward impact of SSW events is highly variable, depending on many factors such as the type of SSW (split vs. displacement; e.g., Charlton and Polvani 2007) or the tropospheric state at the time of the SSW (Domeisen

et al. 2020). Statistically, it is, however, often associated with a strong signal over Greenland, related to Greenland blocking (Domeisen et al., 2020). This is also evident from Fig. 4, showing that the majority of SSW events (red triangles) are followed by a positive geopotential height anomaly over Greenland (i.e., positive GB index).

One of the main goals of this study is to establish a mechanism for the stratospheric influences on MCAOs over the North Atlantic. We assume that changes in the frequency of MCAOs likely arises from modulation of the large-scale flow regime, some of which occurring in connection with stratospheric forcing. For this purpose, the relation between the Greenland blocking and the Scandinavian trough after SSW events has been investigated (Fig. 4). We find that SSW events project on the preferred states for MCAO formation in the Barents and the Norwegian Seas - a positive pressure anomaly over Greenland and a negative anomaly over Scandinavia (Fig. 1e), which have a nearly equivalent contribution to MCAOs in the Barents and the Norwegian Seas (right column in Fig. 5). SSWs therefore contribute to MCAOs in winter primarily through their impact on the Greenland blocking and the Scandinavian trough patterns.

(4) Abstract (L 5-6) says that "Overall, more than a half of SSW events lead to more frequent MCAOs in the Barents Sea." However this is not supported by Figure 2 which shows that exactly half of SSWs are followed by reduced frequency of MCAO in the Barents Sea.

We thank the reviewer for pointing this out. More than half (15 Out of 26 events $\sim$ 58%) of the SSW events in the extended time period (between 1979–2019) are followed by more frequent MCAOs in the Barents Sea compared to the same period in climatology. However, since we do not show this distribution in the revised version, we have rephrased the Abstract accordingly (lines 5-12).

(5) Figure 4 shows that stronger ZD index corresponds to more intense MCAO, likely through more intense northerly flow. But what is the purpose of showing the correlation separately for periods after SSWs? What is the physics behind apparently higher correlation between ZD index and MCAO intensity during periods following SSWs?

We thank the reviewer for this comment. After SSW events, there is evidence of an increased likelihood of a Greenland blocking, as compared to other weather regimes (e.g., Domeisen et al., 2020). As the Greenland blocking contributes to a positive geopotential height anomaly over Greenland (by definition), it makes part of the dipole pattern that is described by the ZDI (see Eq. 2 in the manuscript). The purpose of the correlations shown in Fig. 3 (and in Fig. 4) is to estimate the correlation between ZDI and the MCAOs for each of the sub-regions, and compare between SSW and non-SSW periods. We have rephrased parts of the manuscript to address this point (lines 180-212).

In addition, we added a new figure to the manuscript (Fig. 5) that shows a boxplot distribution of weekly ZDI, $Z'_{GB}$ and $Z'_{ST}$ indices after SSW events and in periods without SSWs. We find that SSW events are associated with a ZDI distribution shift towards positive values, providing an indication of the higher correlation between ZDI and MCAO intensity during periods following SSWs. We added a discussion about these results to the manuscript (lines 213-220).

(6) I am surprised that the authors did not collect the data for the two recent SSW events. Surface impacts by SSWs have a very low signal-to-noise ratio. For example, Maycock and Hitchcock (2015) showed that a large number of events (about 50) are required to detect difference between impacts by splits and displacements. Establishing significant signal seem to be important also for your paper. Adding two recent SSW events would increase the sample size by 8% which is a considerable improvement. The dates for the 2018 and 2019 events could be found for example in the following paper: Afargan-Gerstman, H., Domeisen, D. I. V. (2020). Pacific modulation of the North Atlantic storm track response to sudden stratospheric warming events.Geophysical Research Letters,47,e2019GL085007.

As suggested by the reviewer, we have extended our datasets to 2019 and added the two recent SSW events (2018 and 2019) to this study (see Table 1 in the revised manuscript). Following the reviewers' comments, we have evaluated the significance of the results in several ways: (1) we corrected Fig. 1 to show only statistically significant anomalies at significance level of 95% (panels c-f), (2) we performed additional statistical analysis to evaluate the significance of each correlation in Fig. 3 and Fig. 4, (3) using the Fisher z-est, we assess the significance of the difference between correlations between SSW and non-SSW periods (lines 180-212 and 215-220).

**Other comments:**

L50 After reading the paper I was wondering whether the Barents regions discussed in the paper is so relevant for the densely populated Norwegian coast?

The Barents Sea is important for several reasons. There is a considerable marine activity in the region, mainly due to the oil and gas, fishing, and shipping activities along new trade routes from Asia (Kolstad, 2015). Extreme weather events associated with marine cold air outbreaks, such as polar lows, can cause substantial damage to the marine activity in the Barents Sea region, and pose a risk to the highly-populated Norwegian coast.

L44 "weak stratospheric forcing" Is it a combination of "weak stratospheric vortex" and "stratospheric forcing"?
We have corrected to "weak stratospheric polar vortex" (line 43).

L88-90: How do you calculate MCAO frequency and frequency change after SSW? Please provide equations.

We thank the reviewer for pointing this out, and have added an explanation on calculation of the MCAO frequency to the Methods section. MCAO frequency ($P_M$) is defined as the percentage of days with MCAOs above a 4K threshold within a period of 30 days (see Eq. 1). The climatological MCAO frequency ($P_{M \geq 4K, Clim}$) is computed for all days between December 1 to March 31 divided by the number of days in DJFM. Following SSW events, the MCAO frequency ($P_{M \geq 4K, SSW}$) is computed as the percentage of days with $M \geq 4K$ within a 30-day period after the SSW central date. Hence, the anomaly in the MCAO frequency is the difference between the MCAO frequency in a 30-day period after the SSW central dates and in climatology, as follows

$$\delta P_M = P_{M \geq 4K, SSW} - P_{M \geq 4K, Clim}. \tag{1}$$

To clarify this, we added parts of this paragraph to the revised manuscript (lines 96-103).

L109-110: "Using this classification, we are able to capture the favorable conditions for MCAO occurrence in response to stratospheric forcing." Are these conditions different from those that favor MCAO occurrence without stratospheric forcing?
We find that similar tropospheric conditions contribute to the MCAO occurrence both in climatology and after SSW events. However, these conditions are more likely to arise in the aftermath of SSW events. For a quantitative assessment, Fig. 3 and Fig 4 in the revised manuscript show the regression coefficient for periods after SSW events, compared to the same periods in winters without SSW events.

L191: Should Fig. 5a be replaced by Fig. 4a.
Corrected.

L220: "often followed by a more frequent occurrence" Strange expression. I don't think that saying "It often occurs more frequently" makes much sense but it is what the authors are trying to say.
We thank the reviewer for this comment, and have removed this sentence from the Conclusions.

**Reviewer 2**

**General comments:**

In this manuscript, the authors evaluate whether there is a relationship between marine cold air outbreaks (MCAOs) and Sudden Stratospheric Warmings (SSWs) in the Barents and Norwegian Seas. The authors make the conclusion that 33% of SSWs are associated with an enhanced MCAO response in the Barents Sea. They furthermore conclude that a positive zonal dipole pattern in the large-scale atmospheric flow accounts for 44% of the MCAO variance in the Barents Sea. This manuscript fits within the scope of WCD in that it addresses stratosphere-troposphere coupling, and prediction on subseasonal to seasonal time scales.

The authors present convincing evidence that MCAOs in the North Atlantic are most frequent over the Barents, Norwegian, and Labrador Seas, while MCAOs are more frequent in the Barents and Norwegian Seas in a 30-day period following SSW events. However, I do not think this is strictly a new result (e.g., Fletcher et al. 2016). There is also a convincing case that the Zonal Dipole Index (ZDI) and MCAO are more correlated in the 30-days after an SSW. A key here though is that it is 'more' correlated, and it is not clear what threshold needs to be met in order for there to be a meaningful relationship. Furthermore, the composite patterns after SSWs (Fig. 3) and with MCAOs (Fig. 5) are only roughly similar. Overall, it is my opinion that while this manuscript has some promise, the results are far too premature for publication in WCD at this time.

In particular

1. The term 'enhanced MCAO' refers to "SSW events with an MCAO frequency response above a threshold of 30% are classified as SSWs with a strong MCAO response in the Barents Sea." This 30% is quite arbitrary and no justification is provided and is completely what determines the 33% value in their main conclusion.

Following the comments of the reviewers, we have removed the sub-selection of SSW events based on the strength of the MCAO response. Instead, in the revised manuscript we analyze the influence of all SSW events in the reanalysis. Therefore, no threshold for selection of SSWs is applied throughout the manuscript. Given the significant impact of SSW events on the tropospheric large-scale flow and the associated surface weather (Baldwin and Dunkerton, 2001), analyzing the changes in MCAO frequency and magnitude following all SSW events may shed light on their dominant mechanisms and predictability, with no assumptions or sub-selection made.

2. There is not much of a physical connection with how large-scale fields at 500 hPa and 300 hPa connect to cold air at the surface. Given the frequent elevated inversions in the Arctic, it is not clear under what circumstances the boundary layer couples with the free tropospheric fields. The processes described by Pithan (2018) and Papritz et al. (2019), for example, may be good to incorporate.

In general, MCAOs are most frequently found in the cold sectors of cyclones (Fletcher et al., 2016, Papritz and Grams, 2018, Pithan et al., 2018). The coupling between MCAOs and the large-scale atmospheric flow has been addressed in Papritz and Grams, 2018, showing that the dominant weather regime in the Euro-Atlantic sector essentially modulates the frequency of MCAOs in this region. As they show further, the differences in CAO frequency for each of the weather regimes is closely linked to the modulation of cyclone frequency and tracks in these regimes.

To further address this comment, we have replaced the 300 hPa meridional wind with the 10-m meridional wind (Fig. 2, panels g-i) which represent near-surface winds. For comparison, the large-scale flow is shown by the black arrows in Fig. 2, panels d-f. We added explanation on the coupling between the boundary layer to the Introduction (lines 21-25).

3. To test the relationship between the ZDI and MCAO index, the authors perform a linear regression and show some scatterplots (Fig. 3), arguing that $R2$ is higher just after SSW events compared to climatology. True, the values are higher ($R2 = 0.44$ vs. $R2 = 0.15$ in Fig. 3a), but what value of $R2$ would have made the authors conclude that there is not a difference. $R2 = 0.44$ would not be considered high in many circumstances, so why here?

In the revised manuscript, we have included information on the p-value for each correlation (e.g., lines 187-190). In addition, to obtain an uncertainty estimate for the correlation values, and to test the significance of the diffidence between these correlations, we use the Fisher z-transform. The Fisher z-transform is used to find confidence intervals for r and the differences between correlations. It is commonly used to test the significance of the difference between two correlation coefficients, r1 and r2 from independent samples.

Table 1 below summarizes the results of the statistical correlation test between ZDI and MCAO index in periods that follow SSW events (r1, z'1) and periods without SSWs (r2, z'2) for all three sub-regions of the North Atlantic. The significance of difference between correlations is computed as follows: First, using the z-table we convert the r value to z' score for each correlation. Next, a z-test statistic is constructed ($z'_1 - z'_2$, divided by the standard error). The correlations during SSW and non-SSW periods are considered significantly different at 95% confidence level (i.e., we reject the null hypothesis that they are the same) when the absolute value of $z_{stat}$ is larger than $z_{crit}$. From the z-table, it can be determined that for 95% confidence intervals the critical value is $\pm 1.96$.

We find that in the Barents Sea, the z-test results (bold values in Table 1) indicate that the correlation coefficient between ZDI and MCAO index after SSWs is significantly different than for non-SSW periods. For the Norwegian Sea and the Labrador Sea, however, we cannot reject the null hypothesis that the two correlations are not significantly different at the 95% level. We have added these results to the manuscript (line 187-190).

The results of the significance test for correlations with the $Z'_{GB}$ and $Z'_{ST}$ indices (shown in Table 2) indicate that the difference between SSW and no SSW periods is significant according to a Fisher z-transform for the relation between $Z'_{ST}$ and MCAOs in the Barents Sea. We have added these results to the manuscript (lines 198-201).

Table 1: Correlation coefficients between weekly means of ZDI and MCAO indices for the sub-regions used in this study, during SSW (r1) and non-SSW (r2) periods. Fisher z-transform is used to compute the confidence interval for r1 and r2, and $z_{stat}$ represents the z value for the difference between the two correlations for a two-tailed test with confidence level set to 95%.

| Sub-region | r1 | r2 | z'1 | z'2 | $z_{stat}$ | $z_{crit}$ | |
|---|---|---|---|---|---|---|---|
| Barents Sea | 0.65 | 0.45 | 0.77 | 0.49 | **2.28** | -1.96 | 1.96 |
| Norwegian Sea | 0.55 | 0.49 | 0.63 | 0.54 | **0.68** | -1.96 | 1.96 |
| Labrador Sea | -0.46 | -0.52 | -0.50 | -0.57 | **0.55** | -1.96 | 1.96 |

Table 2: Correlation coefficients between weekly means of $Z'_{GB}$ and MCAO, and $Z'_{ST}$ and MCAO indices for the sub-regions used in this study, during SSW (r1) and non-SSW (r2) periods.

| | Sub-region | r1 | r2 | z'1 | z'2 | $z_{stat}$ |
|---|---|---|---|---|---|---|
| | Barents Sea | 0.34 | 0.26 | 0.35 | 0.27 | **0.70** |
| GB | Norwegian Sea | 0.32 | 0.28 | 0.33 | 0.29 | **0.33** |
| | Labrador Sea | -0.62 | -0.65 | -0.73 | -0.77 | **0.34** |
| | Barents Sea | -0.66 | -0.42 | -0.79 | -0.45 | **-2.67** |
| ST | Norwegian Sea | -0.53 | -0.46 | -0.60 | -0.50 | **-0.77** |
| | Labrador Sea | 0.09 | 0.10 | 0.09 | 0.10 | **-0.02** |

4. There is not convincing evidence that the meridional wind at 300 hPa relates to meridional wind at the surface advecting cold air masses. It is argued that an R2 of 0.13 is more meaningful than an R2 of 0.12 to conclude that there is a stronger relationship between meridional wind and MCAO index in the aftermath of SSW events (line 145). Why not 850 hPa meridional wind (or lower)?

Following the reviewer's comments, we removed this part (i.e., the linear regression analysis between the MCAO index and the meridional wind), and focus on the relation with the geopotential height indices after SSW events and in climatology, which is the main focus on the paper.

5. Table 1 in Butler et al. (2017) provides 24 historical major SSW events between 1979-2014, but the authors analyze atmospheric fields and climatologies from 1979-2016. Thus the range for the analysis period of this study can only be limited through 2014 only. Furthermore, the authors begin with a seasonal December, January, February (DJF) analysis, then switch to also include March (DJFM) midway through (Line 130). They should always use DJFM given that the SSWs contain March events.

Between 2014 and 2016 there were no additional SSW events, thus the analysis period was not limited by using the dates in the Butler et al., 2017 study. Nevertheless, in the revised manuscript we have extended the range of the analysis to 2019, to account for two recent SSW events (in 2018 and 2019). Table 1 in the revised manuscript lists the dates for all SSW events within the study period. Regarding the second point, following the reviewers' suggestion, all results have been updated to an averaging period of DJFM.

6. Why is the climatology not following a standard 30-year climatology as recommended by the WMO (World Meteorological Organization 2017)? Usually it is 1979-2010.

Indeed, the classical period for climatology is 30 years, as defined by the WMO. However, for our purpose, anomalies are calculated with respect to the long-term mean values for each day, and hence the word "climatology" is used to describe this long-term mean (1979–2019).

**Specific comments:**

1. Table 1 in Butler et al. (2017) provides 24 historical major SSW events between 1979-2014, not 2016 as stated. This limits the range for the analysis period of this study to be through 2014 only.
Indeed, Butler et al. (2017) covers the period 1979–2014, whereas 1979–2016 refers to the period used to compute climatology, and therefore this is the full range of the dataset. In the revised version, we have extended the analysis to 2019. The extended period includes two recent SSW events (12th of February 2018 and 2nd of January 2019). In total, 26 SSW events are considered.

2. The analysis corresponding to Figures 1 and 2 is for DJF, when some of the SSW events extend into March as the authors point out (but not until Line 130). Authors should consistently use DJFM instead of strangely adding March 'midstream' on Line 130.

This has already been corrected following the quick reviewers' reports. In both Fig. 1 and Fig. 2 we now analyze DJFM data, consistent with the rest of the manuscript.

3. How are the geographic boxes for the Zonal Dipole Index (ZDI) determined (shown in Fig. 3b)? Justification is needed.

The boxes for the Zonal Dipole Index (ZDI) are determined based on the geographical location of the positive and negative peaks of geopotential height anomalies in a composite of enhanced MCAOs in the Barents Sea (westernmost and easternmost green boxes in Figure 2d in the revised manuscript). We refer to this figure in the Methods section (line 80).

4. Line 50: I assume by 'predictability', the authors actually mean 'practical predictability.' Otherwise elaboration is needed.

Given the increased persistence of surface impacts after SSW events relative to climatology, increased predictability over Europe for medium to long timescales has been proposed after SSW events (Sigmond et al., 2013; Domeisen et al., 2015, 2019; Scaife et al., 2016). A better prediction of MCAOs, as may be achieved after SSW events, would be societally and economically beneficial. To clarify this point, we have rephrased this sentence (line 56) as follows: A better prediction of MCAOs in these regions, due to the increased likelihood of particular surface impacts after SSW events (as compared to climatology) would therefore be societally and economically beneficial.

5. Line 75: ZG and ZE should be ZG and ZE , respectively, given the definition of geopotential height anomaly from climatology in Line 74. Similarly in equation (2).

Thanks for pointing this out, we corrected this.

6. Line 90: Insert 'in the North Atlantic' after 'MCAO index'

Corrected.

7. Line 105: Authors state 'More than half of the SSW events' when I count 12/24 from Figure 2 that are above 25%. Also, the text states that the mean MCAO frequency in DJF is 25%, but in the figure, it looks like 24%. So if it were 25%, that would remove at least one more sample to be less than half.

In the revised manuscript we focus on the downward signature of all SSW events rather than on a sub-selection of certain events. Therefore, we have removed this statement and removed Fig. 2. More specifically, more than half (15 Out of 26 events ∼ 58%) of the SSW events between 1979–2019 are followed by a change in the Barents Sea MCAO frequency that is higher than the mean MCAO frequency during the same period in climatology. Similarly, more than 50% of SSW events exhibit an increase in MCAO frequency in the Norwegian Sea region.

8. Line 120: It is redundant to plot anomalous 500 hPa meridional wind on a 500 hPa geopotential height anomaly plot since flow could reasonably be assumed to be quasi-geostrophic. Furthermore, it is not very convincing to assume that northerly winds at 500 hPa extend to the surface where the cold air outbreak occurs. If the point of this panel is to evaluate the large-scale flow and how it may differ from the average flow during events, it would be more informative to plot the mean 500 hPa height contours instead.

Indeed, as also mentioned by the reviewer the flow at 500 hPa is in near-geostrophic balance, and hence plotting the zonal and the meridional components of the flow pattern is consistent with the geopotential height anomalies at this level. Following this comment, we removed the wind vectors from the geopotential height anomaly panels, and instead added the mean absolute 500-hPa geopotential height (Z500, black contours) for all winter days in ERA-Interim for the reference (Fig. 2, d-f). In addition, we added new panels with the meridional wind at 10-m to provide information on the surface flow during the cold air outbreaks (Fig. 2, g-i).

9. Line 175: The 500-hPa height patterns look quite different between Figures 3b and 5b. After SSWs (Fig. 3b), it the anomalies suggest there is a breaking Rossby wave pattern consistent with LC1 (Thorncroft et al. 1993) with a cut-off low over Central Europe, which is a very different pattern than for the strong MCAOs in the Barents Sea (Fig. 5b). This suggests that on average, the patterns may be considerably different, thus limiting how this relationship could be applied in any prognostic form.

On average, SSWs lead to persistent negative NAO periods (Domeisen, 2019). Changes in the stratospheric conditions (and SSWs in particular) can modulate the weather regimes in the Atlantic–European sector (Charlton-Perez, et al., 2018; Papritz and Grams, 2018), but indeed there is a large event-to-event variability (Domeisen

et al., 2020). While this variability limits the deterministic prediction skill, a statistical connection can be established. As the SSWs contribute to part of the preferred state for enhanced MCAO frequency in the Barents and the Norwegian Seas (i.e., by increasing and decreasing pressure anomalies in the centers of the large-scale zonal dipole pattern), their impact on the ZDI index can be used as a predictor for MCAOs.

To clarify this point, we added an analysis of the different conditions for MCAO occurrence in the different regions (Figure 2d-f, subsection 3.2), highlighting the preferred states for MCAOs in these regions. In addition, we added Fig. 5 to show the large-scale response (represented by the ZDI index, for example) after SSW events, and compared to non SSW periods. As stratospheric precursors such as SSWs often modulate surface weather in the European-North Atlantic regions, their impact on the large-scale circulation pattern contributes to the increased likelihood of MCAOs in these regions after SSWs. We have rephrased the manuscript to include this point (lines 207-224).

10. The text should be reserved to discuss the results of figures, and the caption should provide instructions on how to read and interpret the figure. Lines 115 and 200 are examples where the main text repeats the information in the caption and disrupts the flow of the narrative (Fig. 3b shows..., Fig. 6 shows...).
Following the reviewer's comment, we revised the manuscript to have a better flow and avoid repetitions (for instance, in lines 137-138, 141-142, etc).

11. What are the contours in Figure 3c?
The black contours in Fig. 3c (Fig. 2d-f in the revised manuscript) represent the 500-hPa meridional wind anomaly, and the shading indicates the anomalies which are statistically significant. We added this information to the figure caption.

**Technical corrections:**

1. Line 65: The inline subscripts 'skt' and '850hPa' below equation (1) should be in text mode, consistent with those in equation (1).
Corrected.

2. Lines 75 and 135: ZDI is text, and should be written in text mode, not math mode.
Corrected (ZDI is meant to be written in text mode throughout the manuscript).

3. Line 105: a half → half
Corrected.

4. There should be a space between numbers and units. This occurs frequently with '4K', for example on line 70.
Corrected throughout the manuscript.

**Reviewer 3**

The motivation for the article is worthwhile and will be interesting to readers once a few issues are addressed. In its current form, there are a few gaps in the analysis that need to be filled before publication. I share some concerns with the other two reviewers.

- The first major issues centers on using the top tercile of events as the focus the analysis results. By subsetting of the SSWs into the top tercile of MCAO response, the readers only sees cases that fit one storyline. Fig 2 shows that this MCAO storyline is not always consistent across all SSWs. Since the authors suggest their analysis would inform decision makers that use S2S forecasts, one way to address this issue is by adding analysis of SSW events with non/weak MCAO responses for comparison. Such a solution could involve a corresponding analysis of the bottom tercile events. In such an analysis broader questions could be answered, e.g., are there mechanisms of troposphere-stratosphere coupling that occur in the post-SSW period that favor the enhanced/suppressed MCAO response? Such an analysis would provide readers with needed context for the interpretation of the extreme MCAO response.

To address the reviewers concerns regarding the analysis, we have made several changes to the manuscript, as follows:

- Following the reviewers' comments, we have decided to avoid any sub-selection of SSW events based on a specific storyline, and instead we analyze all SSW events. By doing so, we are able to assess both the mean response of SSWs on the tropospheric and surface flow (Fig. 1d-f), as well as account for their variability (red triangles in Fig. 3 and Fig. 4). We find that SSW events lead to increased MCAOs over the Barents and the Norwegian Seas, while reducing MCAO frequency over the Labrador Sea. In particular, the correlation between the MCAO index and the large-scale tropospheric flow (represented by the ZDI index) is found to be higher after SSW events in the Barents and Norwegian Seas, as compared to climatology, suggesting that the large-scale flow in periods following SSW events favor enhanced MCAO response in these regions. For the Barents Sea, the difference between correlations in SSW and non-SSW periods is found to be statistically significant at the 95% level. We have added a discussion on these results in Section 3.3.

- We have extended the reanalysis datasets to 2019, to account for two additional SSW events. In total, the downward impact of 26 events is considered, and the anomalies of various atmospheric fields, including temperature at 850 hPa, skin temperature (which is later used for the computation of the MCAO index), mean sea level pressure and MCAO frequency, in the period following SSW events are shown (Fig. 1).

- Studying mechanisms of troposphere-stratosphere coupling following SSW events are out of the scope of this paper. However, we investigate the impact of SSW events on the tropospheric large-scale flow in the North Atlantic, in particular the occurrence of Greenland blocking and Scandinavian trough. For this purpose, we analyze the relation between the large-scale tropospheric flow and MCAOs in climatology and after SSW events (Fig. 3 and Fig. 4). We find that SSW events are often associated with enhanced blocking over Greenland and lower pressure anomaly over central Europe and Scandinavia, which projects on the preferred geopotential height anomaly for MCAOs in the Barents and Norwegian Sea (and the opposite for the Labrador Sea MCAOs). Through this signature, and its associated surface circulation, there is an increased likelihood of MCAOs in these regions after SSW events (see discussion on correlation coefficients in Fig. 3 and Fig. 4).

- Adding an analysis along these lines to the article would also get at addressing a second major issue, the connection between the stratosphere and the large-scale tropospheric flow is assumed and not shown as part of this work in its present form. Is there a stratospheric flow metric that would provide an indication of the likelihood of whether or not the 30-day period after the SSW would have a bottom or top tercile event or is it not possible to determine at the SSW onset date? With the addition of such an analysis, the authors could potentially provide insight into the type/evolution of SSW events required for the high-impact MCAO response.

Generally, nearly two thirds of SSW events are followed by the canonical downward response in the North Atlantic, characterized by a dominant negative NAO pattern and an equatorward shift of the North Atlantic tropospheric jet (e.g., Afargan-Gerstman and Domeisen, 2020). There is an extensive body of literature relating the type and strength of the stratospheric anomaly during an SSW to the tropospheric impact. We think it will be beyond the scope of this paper to explore this, as our focus is on the tropospheric MCAO-related impact. Given the small number of SSW events in the record, and as pointed out by the reviewers, we think that sub-sampling the SSW events will yield insignificant results, e.g. with respect to the stratospheric evolution. Furthermore, we think that it will be a complex task, or not possible, to relate the type of MCAO impact at the surface to the stratospheric forcing at the time of occurrence of the SSW event.

**References:**

- Afargan-Gerstman, H., Domeisen, D. I. (2020). Pacific modulation of the North Atlantic storm track response to sudden stratospheric warming events. Geophysical Research Letters, 47(2), e2019GL085007.

- Butler, A. H., J. P. Sjoberg, D. J. Seidel, and K. H. Rosenlof, 2017: A sudden stratospheric warming compendium. Earth System Science Data, 9.

- Charlton, A. and Polvani, L., 2007: A new look at stratospheric sudden warmings. Part I: Climatology and modeling benchmarks. J. Climate, 20.

- Domeisen, D. I. (2019). Estimating the frequency of sudden stratospheric warming events from surface observations of the North Atlantic Oscillation. Journal of Geophysical Research: Atmospheres, 124(6), 3180-3194.

- Domeisen, D. I., Grams, C. M., Papritz, L. (2020). The role of North Atlantic-European weather regimes in the surface impact of sudden stratospheric warming events. Weather and Climate Dynamics Discussions, 1-24.

- Fletcher, J., S. Mason, and C. Jakob, 2016: The climatology, meteorology, and boundary layer structure of marine cold air outbreaks in both hemispheres. J. Climate, 29 (6), 1999-2014.

- Kolstad, E. W. (2015). Extreme small-scale wind episodes over the Barents Sea: When, where and why?. Climate Dynamics, 45(7-8), 2137-2150.

- Maycock, A. C., and P. Hitchcock (2015), Do split and displacement sudden stratospheric warmings have different annular mode signatures?, Geophys. Res. Lett., 42, 10,943–10,951, doi:10.1002/2015GL066754.

- Papritz, L., Grams, C. M. (2018). Linking low-frequency large-scale circulation patterns to cold air outbreak formation in the northeastern North Atlantic. Geophysical Research Letters, 45(5), 2542-2553.

- Papritz, L., E. Rouges, F. Aemisegger, and H. Wernli, 2019: On the thermodynamic pre-conditioning of arctic air masses and the role of tropopause polar vortices for cold air outbreaks from Fram strait. J. Geophys. Res.: Atmos.

- Pithan, F., et al., 2018: Role of air-mass transformations in exchange between the Arctic and mid-latitudes. Nat. Geosci., 11 (11), 805.

- Scaife, A. A., Knight, J. R., Vallis, G. K., Folland, C. K. (2005). A stratospheric influence on the winter NAO and North Atlantic surface climate. Geophysical Research Letters, 32(18).

- Sigmond, M., Scinocca, J. F., Kharin, V. V., Shepherd, T. G. (2013). Enhanced seasonal forecast skill following stratospheric sudden warmings. Nature Geoscience, 6(2), 98-102.

- Thorncroft, C. D., B. J. Hoskins, and M. E. McIntyre, 1993: Two paradigms of baroclinic-wave life-cycle behaviour. Quart. J. Roy. Meteor. Soc., 119, 17-55.

- World Meteorological Organization, 2017: WMO guidelines on the calculation of climate normals. World Meteorological Organization Switzerland.

---

## Author Response (AR2)

**Response to Reviewers**

We would like to thank the reviewers for their positive evaluation and helpful comments on our study. Please find below a point-by-point reply (in blue) to the reviewers' comments and suggestions. All changes are highlighted in the attached version of the manuscript.

With kind regards, Hilla Afargan-Gerstman and co-authors

**Comments from the Editor**

1. In Fig. 2 you may want to only highlight the relevant boxes in the resp. panels (e.g., only highlight the Barents Sea box in the left column etc)

We have revised Fig.2 as suggested, such that only the relevant box in the respective panels is highlighted.

2. Please indicate how you obtain the (effective) d.o.f. for the confidence intervals that you computed for the Fisher z-test. In particular, did you take into account autocorrelation of the weekly indices, which will reduce the effective d.o.f.?

The Fisher z-test is used in the manuscript for determining the significance of the difference between SSW and non-SSW correlations. Generally, the decorrelation timescale in the North Atlantic has been estimated as 5 to 7 days for the NAO (depending on the dataset used, e.g., Domeisen et al., 2018) and around 6 to 15 days for the NAM, with a peak in mid-winter (Gerber et al., 2008). We compute the decorrelation timescale for air temperature at 850 hPa averaged over the Barents Sea region for daily and weekly means (Fig. 1 below). The decorrelation timescale (T) is estimated by the time interval over which the autocorrelation drops to 1/e. We find that T drops to 1/e after 4 days for the daily data, and after 1 week for the weekly data. Based on these results, we estimate that there is no autocorrelation between weekly indices beyond 1 week. Thus, by using weekly means, rather than a 7-day running average, we account for any autocorrelation below 1 week.

Figure 1: The autocorrelation function of lag time for 850-hPa temperature over the Barents Sea region in DJFM using (left) daily and (right) weekly means. Data is based on ERA-Interim reanalysis (1979–2019). The dashed line is used for estimation of the time lag over which the autocorrelation drops to 1/e.

The number of degrees of freedom relative to the number of observations can be estimated as half the efolding time (i.e., the time for one-lag autocorrelation to reduce to 1/e). Thus, we use one half of the number of observations as the effective degree of freedom for no SSW periods (n2=276). For the periods following SSW events, we assume the 26 SSW events are independent of each other. Table 1 and Table 2 below shows the outcome of using 1-week and 2-weeks as the decorrelation time after SSW events, respectively. The estimation of n1 is taken as half that time, i.e., 26x2 and 26x4.

In both cases, the difference between SSW and no SSW periods is found to be significant at the 95% level for the Barents Sea (p<0.05), whereas for the Norwegian and Labrador Seas the difference is found to be significant

Table 1: Correlation coefficients between weekly means of ZDI and MCAO indices for the sub-regions used in this study, during SSW (r1) and no SSW (r2) periods. Fisher z-transform is used to compute the confidence interval for r1 and r2.  $z_{stat}$  represents the z score for the difference between the two correlations, divided by the standard error. The values of n1 and n2 correspond to the d.o.f used in the computation of the standard error. The z'1 values in the table were computed using a 1-week decorrelation time after SSW events.

| Sub-region    | r1    | r2    | n1   | n2  | z'1   | z'2   | Z stat | p-value |
|---------------|-------|-------|------|-----|-------|-------|-------------------|---------|
| Barents Sea   | 0.65  | 0.45  | 26x2 | 276 | 0.77  | 0.49  | 1.87              | 0.03    |
| Norwegian Sea | 0.55  | 0.49  | 26x2 | 276 | 0.63  | 0.54  | 0.53              | 0.30    |
| Labrador Sea  | -0.46 | -0.52 | 26x2 | 276 | -0.50 | -0.57 | 0.51              | 0.30    |

Table 2: Same as in Table 1, but using 2-weeks as the decorrelation time after SSW events.

| Sub-region    | r1    | r2    | n1   | n2  | z'1   | z'2   | $z_{stat}$ | p-value |
|---------------|-------|-------|------|-----|-------|-------|------------|---------|
| Barents Sea   | 0.65  | 0.45  | 26x4 | 276 | 0.77  | 0.49  | 2.50       | 0.006   |
| Norwegian Sea | 0.55  | 0.49  | 26x4 | 276 | 0.63  | 0.54  | 0.71       | 0.24    |
| Labrador Sea  | -0.46 | -0.52 | 26x4 | 276 | -0.50 | -0.57 | 0.68       | 0.25    |

only at  ${\sim}70\%$  for one week decorrelation time, and 75% for two weeks.

To clarify this point in the revised manuscript, we added a statement that the autocorrelation of the weekly indices has been properly accounted for in estimating the degree of freedom (line 195). Particularly, we mention that the Fisher z-test is based on an estimation that there is no autocorrelation between weekly indices after a period of one week in DFJM. Assuming a longer decorrelation times after SSWs (e.g., two weeks) leads to a qualitatively similar conclusion.

**Reviewer 1**

**General comments:**

In this manuscript, the authors are evaluating whether there is a relationship between marine cold air outbreaks (MCAOs) and Sudden Stratospheric Warmings (SSWs) in the Barents, Norwegian, and Labrador Seas. The authors conclude that changes in the large-scale tropospheric circulation account for 42% of the MCAO variance in the Barents Sea and 31% of the variance in the Norwegian Sea. They also make a convincing case that there is a significant increase in the correlation between the large-scale tropospheric flow pattern and MCAOs after SSWs in the Barents and Norwegian Seas. The connection to the large-scale tropospheric flow to MCAOs is further found to be correlated with the Scandinavian Trough pattern in the Barents and Norwegian Seas and Greenland Blocking pattern in the Labrador Sea. This manuscript fits within the scope of WCD in that it addresses stratosphere-troposphere coupling and prediction on sub-seasonal to seasonal time scales. In this revised manuscript, the authors have diligently addressed the earlier concerns and I now think that this could be published after some minor revisions outlined below.

**Specific comments:**

1 The connection to physical processes linking the stratosphere to the surface has been improved in this version, although after reading again, I am still left wondering what physical feature(s) is(are) transporting the cold air into the regions of focus in this study. It makes sense that the lower 500 hPa heights would be associated with the colder air and the related storminess nearby. Papritz et al. (2019) address these processes that are likely relevant in the Norwegian Sea, particularly the tropopause polar vortex (e.g. Cavallo and Hakim 2010).

Changes in the frequency of MCAOs in the North Atlantic have been suggested to arise in the first place from shifts in the strength and the location of cyclone activity. Fletcher et al. (2016) showed that MCAOs occur predominantly in the cold sector of cyclones. This has been further detailed by Papritz and Grams (2018) for the Nordic Seas and the Barents Sea. By considering cyclonic and anticyclonic flow regimes, they demonstrated this relationship showing that the MCAO frequency is enhanced west of positive cyclone frequency anomalies and MCAO occurrence is suppressed in regimes with no or weak cyclone activity across the Nordic Seas and the Barents Sea. In particular, intense storm track activity associated with cyclonic weather regimes is found to be favorable for MCAOs in these regions. Hence, the stratospheric impact on MCAOs is thought to modulate the large-scale flow as reflected in 500 hPa geopotential height, which in turn affects the tracks and frequency of cyclones. The anomalous near surface flow associated with changes to the cyclone tracks then drives changes in the advection of cold air masses and, thus, in the occurrence of MCAOs.

To clarify this point in the manuscript, we have revised part of the Introduction and added a discussion on the physical processes that affect the variability of MCAOs in the northeast North Atlantic (lines 23–35). Additionally, we have revised the order of the panels in Fig. 2 and extended the discussion on cyclone activity as one of the main drivers for MCAOs in these regions.

Another process that is likely relevant for the variability of MCAOs in these regions is the occurrence of cyclonic tropopause polar vortices (e.g. Cavallo and Hakim 2010; Papritz et al. 2019). These vortices can induce especially intense MCAOs over the Nordic Seas when they propagate out of the high Arctic (Papritz et al. 2019). It would be interesting to investigate whether tropopause polar vortices are more or less likely to propagate southward in the aftermath of an SSW. Due to the low frequency of SSWs and the relatively low frequency of tropopause polar vortices, a statistical analysis of this mechanism is unfortunately not possible based on the reanalysis.

2 Line 50: The Labrador Sea is also mentioned in the abstract and on line 55, but just the Norwegian and Barents Seas here.

We added a new sentence on the Labrador Sea (line 60), mentioning that the air-sea fluxes associated with MCAOs may have an impact on dense water formation in this region, and hence on the North Atlantic overturning circulation.

3 Line 129: Please provide a justification for choosing M  $\ge$  4K to define moderate-to-strong MCAOs in DJFM climatology.

A separation to weak (< 4 K), moderate (4 to 8 K) and strong (> 8 K) MCAOs by defining thresholds for the MCAO index (M) can be found in Papritz and Spengler (2017). Using these categories, we identify moderate-to-strong MCAOs with a threshold of M >= 4 K (i.e., a combination of moderate and strong categories). As pointed out by Papritz and Spengler (2017), MCAOs with an intensity of in excess of 4 K are associated with notable upward heat fluxes from the ocean to the atmosphere. To clarify this point, we added the following sentence to the manuscript (lines 80-81): "In this study we focus on MCAOs with an MCAO index (Eq. 1) exceeding a threshold of 4 K. This choice of threshold is in accordance with the thresholds used in Papritz and Spengler (2017) and selects moderate-to-strong MCAOs with notable upward heat fluxes from the ocean. Other studies have used slightly different thresholds for the MCAO index (e.g., a threshold of 3 K for moderate MCAO events in Fletcher et al. (2016)) however the results presented in this study are not sensitive to small changes of this threshold (of order of 1-2 K)".

4 Lines 135-160: The interpretation provided that the storminess increases in the Barents, Norwegian and Labrador Seas at the time of MCAOs in the respective regions is not quite accurate. However, the conclusion reached by the authors on lines 142-144 is the correct conclusion. Cold air outbreaks occur when there are northerly winds over those regions, which means that the storms must move slightly east of those regions so that the cold sector (west side) is over the seas themselves. For example, in Figure 2j, there is an increase in storminess in the lower right side of the Barents Sea box but decrease in upper left side (not stated correctly on lines 137-138) and similarly for the Norwegian Sea (Fig. 2k) and Labrador Sea (Fig. 2l). This also applies to the conclusions section on line 265.

We agree with the reviewer's view, that cold air outbreaks occur when cold air masses from the ice-covered polar areas are transported over the ocean, as it typically occurs in the cold sectors of storms. Following this comment, we have revised parts of the Results section (lines 135–160) accordingly. We have also revised the Conclusion section in accordance with this perspective (lines 270–275). We emphasize that the occurrence of MCAOs in each of the regions is associated with increased cyclone frequency east of the domain.

5 Line 140: On point (iii), it looks like in Figure 2g that the strongest northerly wind anomalies are over the Barents Sea with weaker northerly wind anomalies over the Norwegian Sea.

Following the reviewer's comment, we have changed this sentence (line 150) to clarify that strong northerly winds in periods of enhanced MCAOs in the Barents Sea are found over the Norwegian and the Barents Sea in particular (Fig. 2j).

6 Lines 180-190: It is nice that the authors have now considered statistical significance in this revision. However, it is not clear on line 187 what "All correlations...are found to be statistically significant" refers to. It is good that the correlations themselves have p < 0.05, but what about the differences in correlations between the SSW and climatology or SSW and no SSW? This seems to be addressed in the Labrador Sea correlations only. It should be stated which correlations are and are not significant; perhaps a table with the corresponding p-values would be most helpful.

Following this comment, we have clarified in the manuscript that all correlation coefficients are significant at the 95% level (a statement was added to Fig. 3 and Fig. 4 caption). In addition, in the revised version we included more information of the significance of the difference between SSW and non-SSW correlations computed using Fisher's z-test (lines 194-200). We find that only in the Barents Sea region the differences in weekly ZDI-MCAO correlations between SSW and non-SSW periods is significant at 95%, whereas in the Norwegian and Labrador Sea regions this difference is significant at a lower level of 75%.

The significance of the differences between the GB-MCAO and ST-MCAO correlations for SSW and non-SSW periods is also investigated. The result of this statistical test is summarized in the manuscript as follows (lines 212-214): A significance difference at a 95% level between SSW and non-SSW periods is found only the ST-MCAO relation in the Barents Sea region.

7 Line 209: Where are the authors getting that there is an increase in 15% in the explained variance for the Barents Sea for the ZDI index vs. MCAO? It looks like it should be 18% from Figure 3a.

Corrected. We changed the value of the explain variance to 18% in accordance with the values in Fig. 3a.

8 Line 245: It is easier to see from Figure 6a,b that the circulation over the Barents Sea is "anomalously cyclonic" rather than "cyclonic."

We replaced "cyclonic" to "anomalously cyclonic" as suggested (line 247). We have also rephrased lines 245–250 accordingly.

**Technical corrections:**

- Figure 2: The caption should state what the black and green boxes are in each of the panels (as it is in Figure 1 caption).
  Corrected. We added a description of the boxes to Figure 2 caption.
- 2. It is not necessary to say 'historical' on line 90 Corrected, replaced this term with "observed".
- 3. Line 145: enhanced -> moderate-to-strong Corrected, also in line 150 (i.e., we show the MCAO anomalies for periods of moderate-to-strong MCAOs in each region).

**Reviewer 2**

The manuscript has improved substentially after the revision. Thus, I recommend it for publication. Please also check the following comment: L118: Should not it be eastern North Atlantic?

We thank the reviewer for their positive evaluation. We corrected to "northeastern North Atlantic" on this line and throughout the manuscript.

**Reviewer 3**

The authors have done a good job of addressing the comments from the first round of review. Their effort has greatly improved the manuscript. I think by both broadening the analysis to include all SSW events rather than just a limited subset from the reanalysis period, as well as, including the additional North Atlantic regions into the analysis, the paper provides a compelling synoptic climatological analysis on the occurrence of anomalous MCAO activity after SSW. I have a few minor/technical comments about the framing of the analysis and have included these below.

**Minor comments:**

1. Title change: The title needs to be more specific to the analysis presented in the paper. The authors only consider sudden stratospheric warming events and not other 'stratospheric influences' like those from final warming events, strong events, etc. Perhaps something like, "Modulation of North Atlantic marine cold air outbreaks following sudden stratospheric warming events" would be more appropriate to the analysis presented.

We thank the reviewer for this suggestion. We have changed the title of the paper accordingly to "Stratospheric influence on North Atlantic marine cold air outbreaks following sudden stratospheric warming events".

2. The authors state that they "aim to shed light on the predictability of MCAOs" on line 58 and reemphasize this point elsewhere in the discussion of the results. However, this aim should be reworded/scaled back. The analysis does not get into a discussion about our ability to predict MCAO at various timescales (nor should it, as that is not the focus of the analysis), rather shows that MCAOs occur more frequently in some locations and less so in others after SSWs. Perhaps this fact could be used by a simple statistical model to increase predictability, but that is not explicitly discussed. This aim should be reframed, perhaps in terms of mitigating societal/economic impacts (i.e., those discussed in the intro) due to changes in frequency and locations.

Indeed, the ability to predict MCAOs at various timescales is not the focus of this study, which focuses on the influence of large-scale atmospheric patterns in the northeastern North Atlantic on MCAOs in these regions, and their link to sudden stratospheric warmings.

To better address this issue, we have revised several paragraphs in the manuscript. In particular, in the Introduction (lines 62–64) we emphasize that our results aim to shed light on the precursors and occurrence of MCAOs over the Barents and Norwegian Seas, as well as for the Labrador Sea, which is expected to benefit long-range predictability of their extreme impacts. Given the long-lasting influence of SSW events on the tropospheric circulation on mid- and high-latitudes, this connection can potentially be exploited for improving subseasonal MCAO predictions. In addition, by performing linear regression, we have been able to demonstrate a statistical relationship between MCAOs and other atmospheric indices that capture the characteristics of the large-scale flow. Such a connection can be further used for mitigation of societal and economic impacts by providing an estimate of the increase/decrease in MCAO intensity due to a change in the environmental conditions.

3. Fig. 2: A minor suggestion to consider: perhaps you can flip your color bar for the first row of the figure (for MACO) so that cold air is represented by cold colors and warm air is warm colors.

We thank the reviewer for this suggestion. We agree that using cold color scheme for the cold air makes sense in this context, however we prefer to visually distinguish the MCAO anomaly from the northerly wind flow in the lower panels of Fig. 2, and therefore we choose warm colors to demonstrate the enhanced MCAO intensity, rather than cold colors.

**References**

- Cavallo, S. M. and G. J. Hakim, 2010: The composite structure of tropopause polar cyclones from a mesoscale model. Mon. Wea. Rev., 138 (10), 3840-3857, doi:10.1175/2010MWR3371.1.
- Fletcher, J., Mason, S., Jakob, C. (2016). The climatology, meteorology, and boundary layer structure of marine cold air outbreaks in both hemispheres. Journal of Climate, 29(6), 1999-2014.
- Domeisen, D. I. V., Badin, G., Koszalka, I. M. (2018). How Predictable Are the Arctic and North Atlantic Oscillations? Exploring the Variability and Predictability of the Northern Hemisphere. Journal of Climate, 31(3), 997–1014.
- Gerber, E. P., Polvani, L. M., Ancukiewicz, D. (2008). Annular mode time scales in the intergovernmental panel on climate change fourth assessment report models. Geophysical Research Letters, 35(22).
- Papritz, L., Grams, C. M. (2018). Linking low-frequency large-scale circulation patterns to cold air outbreak formation in the northeastern North Atlantic. Geophysical Research Letters, 45(5), 2542-2553.
- Papritz, L., Spengler, T. (2017). A Lagrangian climatology of wintertime cold air outbreaks in the Irminger and Nordic Seas and their role in shaping air-sea heat fluxes. Journal of Climate, 30(8), 2717-2737.

• Papritz, L., E. Rouges, F. Aemisegger, and H. Wernli, 2019: On the thermodynamic pre-conditioning of arctic air masses and the role of tropopause polar vortices for cold air outbreaks from Fram Strait. J. Geophys. Res.: Atmos.